# SSDM: Scalable Speech Dysfluency Modeling

Jiachen Lian[1], Xuanru Zhou[2], Zoe Ezzes[3], Jet Vonk[3], Brittany Morin[3], David Baquirin[3], Zachary Miller[3], Maria Luisa Gorno Tempini[3], Gopala Anumanchipalli[1]

[1] UC Berkeley, [2] Zhejiang University, [3] UCSF
{jiachenlian, gopala}@berkeley.edu

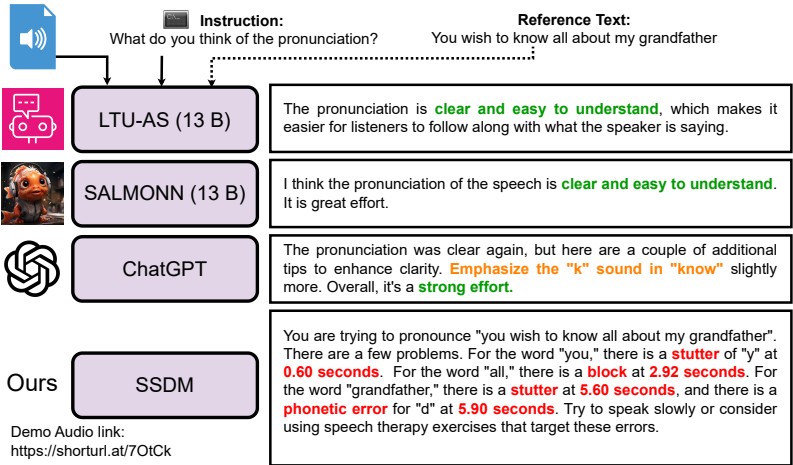

Figure 1: SSDM. Comparison to other methods
.

## Abstract

Speech dysfluency modeling is the core module for spoken language learning, and speech therapy. However, there are three challenges. First, current state-of-the-art solutions [1, 2] suffer from poor scalability. Second, there is a lack of a large-scale dysfluency corpus. Third, there is not an effective learning framework. In this paper, we propose *SSDM: Scalable Speech Dysfluency Modeling*, which (1) adopts articulatory gestures as scalable forced alignment; (2) introduces connectionist subsequence aligner (CSA) to achieve dysfluency alignment; (3) introduces a large-scale simulated dysfluency corpus called Libri-Dys; and (4) develops an end-to-end system by leveraging the power of large language models (LLMs). We expect SSDM to serve as a standard in the area of dysfluency modeling. Demo is available at `https://berkeley-speech-group.github.io/SSDM/`.

## 1 Introduction

Speech dysfluency modeling is key for diagnosing speech disorders, supporting language learning, and enhancing therapy [1]. In the U.S., over 2 million individuals live with aphasia [3], while globally, dyslexia affects approximately one in ten people [4]. The U.S. speech therapy market is projected to reach USD 6.93 billion by 2030 [5]. This growth *parallels* developments in Automatic Speech Recognition (ASR), valued at USD 12.62 billion in 2023 [6], and Text-to-Speech (TTS), valued at USD 3.45 billion [7]. Moreover, the global language learning market is anticipated to be USD 337.2 billion by 2032 [8]. Conversely, substantial investments have been made in training speech-language pathologists (SLPs) [9, 10], and the high cost of treatment often remains out of

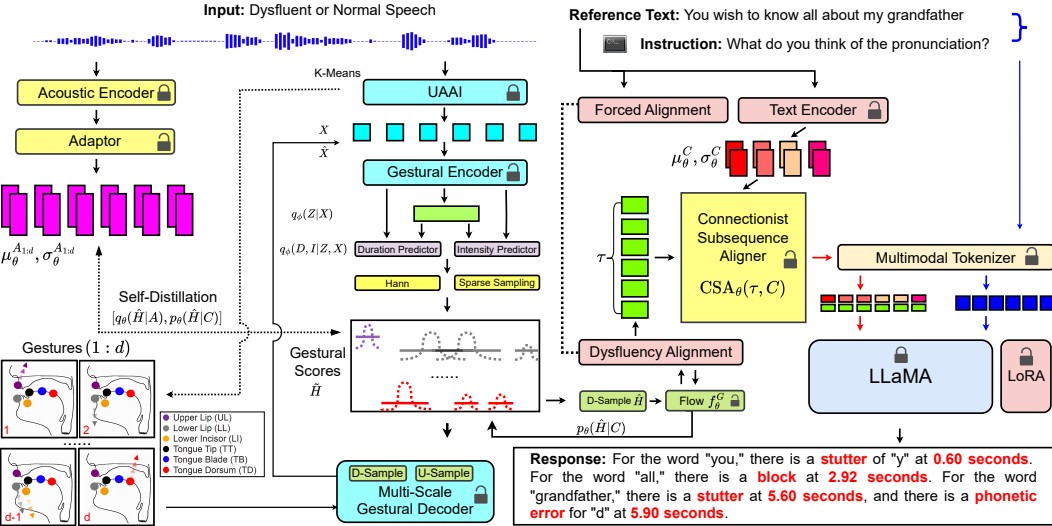

Figure 2: SSDM architecture

reach for many low-income families [11–15]. Therefore, there is a crucial need for an AI solution that makes advanced speech therapy and language learning *available and affordable for everyone*.

Speech dysfluency modeling detects various dysfluencies (stuttering, replacements, insertions, deletions, etc) at both word and phoneme levels, with accurate timing and typically using a reference text [1]. (see Figs.1 for examples). Fundamentally, it is a *spoken language understanding problem*. Recent advancements have been driven by large-scale developments [16–31]. However, these efforts often focus on scaling coarse-grained performance metrics rather than deeply listening to and understanding the nuances of human speech.

Traditional approaches to dysfluency modeling have relied on hand-crafted features [32–36]. Recent advancements have introduced end-to-end classification tasks at both utterance [37–48] and frame levels [49, 50]. However, these methods often overlook internal dysfluency features like alignment [1] and struggle to detect and localize multiple dysfluencies within a single utterance. [1, 2] propose 2D-Alignment, a non-monotonic approach that effectively encodes dysfluency type and timing. Nonetheless, initial experiments show that this method struggles with scalability, limiting its further development. To address these concerns, we revisit this problem and summarize our contributions as follows:

- We revisit speech representation learning from a physical perspective and propose *neural articulatory gestural scores*, discovered to be scalable representations for dysfluency modeling.
- We introduce the *Connectionist Subsequence Aligner* (CSA), a differentiable and stochastic forced aligner that links acoustic representations and text with dysfluency-aware alignment.
- We enable *end-to-end* learning by leveraging the power of large language models.
- We open-source the large-scale simulated dataset *Libri-Dys* to facilitate further research.

## 2 Articulatory Gesture is Scalable Forced Aligner

### 2.1 Background

**Revisit Speech Representation Learning** Self-supervised speech representations [51], large-scale ASR [16–18, 20], codec models [52–67], and speech language models (SLMs) [21–31] have emerged as universal paradigms across tasks and languages. However, high computing costs of scaling efforts is not affordable for academia researchers. In this work, we propose learning speech representations grounded in fundamental physical laws [68, 69]. This approach characterizes speech representations by the kinematic patterns of articulatory movements, a method we refer to as *gestural modeling*.

**Gestural Modeling**  The concept of *gesture*, as defined by [70, 71], refers to articulatory movements in acoustic space, similar to body gestures in humans. [70, 71] introduced *gestures* as a dictionary of basic articulatory movements and *gestural scores*, representing the *duration* and *intensity* of these movements. This principle resembles the gait library and optimization used in robotics [72]. The computational modeling of gestures was first developed by [73], using sparse matrix factorization [74, 75] to decompose EMA data [76] into interpretable components. Further research by [77] and [78] streamlined this into an end-to-end neural approach. *Gestural scores* serve as speech representations. We *discovered* that they serve as *scalable dysfluent phonetic forced aligner*.

**Scalable Dysfluent Phonetic Forced Aligner**  Dysfluency modeling requires detecting both the type and timing of dysfluencies, necessitating the use of forced alignment [1]. This alignment is often non-monotonic (e.g., stuttering). Thus, previous monotonic alignment methods [79, 80, 20, 81] perform poorly in the dysfluency domain. The primary challenge is the inherent uncertainty in what the speaker actually said, compounded by invariably inaccurate reference texts, as explained in [1]. Effective research in this area focuses on non-monotonic alignment modeling. [82] introduces the WFST [83] to capture dysfluencies such as sound repetition. However, it assumes the actual speech does not deviate significantly from the reference text. [1] proposed *2D-alignment* as final dysfluent representation. Nevertheless, this method, and its extension [2], suffers from scalability issues: *increasing training data does not lead to further improvements*. In this work, we revisit the monotonic alignment to tackle the scalability problem. To achieve this, we need a scalable representation, and a scalable monotonic aligner (Sec. 3). This section focuses on the first part and proposes *Neural Variational Gestural Modeling* to deliver *gestural scores $H$* as scalable dysfluent speech representations. We also provide a visualization of *gestures* and *gestural scores* in Appendix. A.1.

## 2.2  Neural Variational Gestural modeling

Despite theoretical support [70, 71, 68, 69], gestural scores have not yet become a universal speech representation [51] due to several limitations. First, gestural modeling requires extensive, often unavailable, articulatory kinematic data. Second, there is not an effective learning framework. Third, the commonly used EMA data, sampled sparsely from human articulators [84–87], suffer from information loss. To overcome these challenges, we proposed *Neural Variational Gestural Modeling*. This model uses an offline inversion module (Sec. 2.2.1) to capture articulatory data, and a gestural VAE to extract gestural scores (Sec. 2.2.2), which are then refined through joint self-distillation with acoustic posteriors and textual priors (Sec. 2.2.3). This method ensures that the resulting gestural scores are effective and scalable dysfluent speech representation. (Evidenced in Sec. 6)

### 2.2.1  Universal Acoustic to Articulatory Inversion (UAAI)

Since the real articulatory data are typically unavailable, we employ a state-of-the-art acoustic-to-articulatory inversion (AAI) model [88] pretrained on MNGU0 [84]. The model takes 16kHz raw waveform input and predicts 50Hz EMA features. Details are listed in Appendix. A.2.1.

### 2.2.2  Gestural Variational Autoencoders

Any motion data $X = [X_1, X_2, ..., X_t]$ can be decomposed into motion kernels $G \in \mathbb{R}^{T \times d \times K}$ and an activation function $H \in \mathbb{R}^{K \times t}$ using convolutional matrix factorization (CMF) [75], where $X \approx \Sigma_{i=0}^{T-1} G(i) \cdot \overrightarrow{H}^i$. Here, $t$ represents time, $T$ the kernel window size, $d$ the channel size, and $K$ the number of kernels. When $X$ is articulatory data, $G$ corresponds to $K$ gestures and $H$ to the gestural scores (Visualization in Appendix A.1 and A.1.2). This work focuses on three aspects: (1) joint modeling of articulatory-specific *duration and intensity*, (2) *self-distillation* from both acoustic and textual data, and (3) *multi-scale* decoding of gestures and gestural scores.

**Variational Inference**  We employ point-level variational inference for $q_\theta(H|X)$, meaning for each point $(k,i)$ in $H \in \mathbb{R}^{K \times t}$, we model its posterior $q_\theta(H^{k,i}|X)$. This approach results in $K \times t$ posteriors for each gestural score $H$, where $k = 1, \ldots, K$ and $i = 1, \ldots, t$. We use pointwise inference for gestural scores due to its properties, such as overlapping durations across articulators and stochastic variations across accents. We will refer to this as *patchwise* rather than pointwise, as we are modeling a patch embedding for each point $(k,i)$. In practice, we introduce an additional latent vector $Z^{k,i} \in \mathbb{R}^P$ as variational augmentation [89], where $P$ is patch size. This setup formulates the duration posterior $q_\phi(D^{k,i}|Z^{k,i}, X)$, intensity posterior $q_\phi(I^{k,i}|Z^{k,i}, X^{k,i})$, and latent posterior $q_\phi(Z^{k,i}|X)$. Patchwise operation is detailed in Appendix A.2.2. Consequently, our gestural encoder encodes the joint posterior $q_\phi(Z^{k,i}, D^{k,i}, I^{k,i}|X) = q_\phi(D^{k,i}|Z^{k,i}, X)q_\phi(I^{k,i}|Z^{k,i}, X^{k,i})q_\phi(Z^{k,i}|X)$.

**VAE Objective**  After variational inference, our decoder $p_\theta(X|H, G) = P_\theta(X|D, I, G)$ reconstructs $X$ using duration $D$, intensity $I$, and gesture $G$. The evidence lower bound (ELBO) and its derivation are provided in Eq. 1 and Appendix A.4, respectively. The posterior $q_\phi(Z^{k,i}|X)$, modeled via vanilla variational inference [90], assumes standard normal priors for $p(Z^{k,i})$. The mechanisms of the duration and intensity encoders, $q_\phi(D^{k,i}|Z^{k,i}, X^{k,i})$ and $q_\phi(I^{k,i}|Z^{k,i}, X^{k,i})$, are detailed in Sec. 2.2.2 and Sec. 2.2.2. Details on the decoder $P_\theta(X|D, I, G)$ are discussed in Sec. 2.2.2.

$$\mathcal{L}_{\text{ELBO}} = \mathbb{E}_{q_\phi(Z,D,I|X)}\left[\log p_\theta(X|D, I, G)\right]$$
$$- \mathbb{E}_{(k,i)\sim\mathbb{S}}\left[\text{KL}\left(q_\phi(Z^{k,i}, D^{k,i}, I^{k,i}|X)\|p(Z^{k,i}, D^{k,i}, I^{k,i}))\right)\right] \quad (1)$$

**Duration Posterior** $q_\phi(D^{k,i}|Z^{k,i}, X^{k,i})$  We employ the Gumbel softmax [91] to reformulate the duration posterior $q_\phi(D^{k,i}|Z^{k,i}, X)$. Let $\pi^{k,i} \in \mathbb{R}^{\mathbb{C}}$ denote the logits across all $\mathbb{C}$ discrete duration classes (values) for patch $(k, i)$. For each class $j$, we obtain Gumbel noise $\epsilon_j^{k,i} = -\log(-\log(U_j))$, where $U_j \sim \text{Uniform}(0, 1)$. We then define $\tilde{\pi}_j^{k,i} = (\log(\pi_j^{k,i}) + \epsilon_j^{k,i})/\tau$, where $\tau$ is temperature parameter. Finally, we obtain the Gumbel softmax transformation as an approximation of the duration posterior in Eq.2. We set $p(D^{k,i}) = 1/\mathbb{C}$, where $\mathbb{C}$ is the number of discrete duration classes. Background and detailed methodology can be viewed in Appendix. A.2.2.

$$q_\phi(D^{k,i}{=}j|Z^{k,i}, X) \approx \frac{\exp\left(\tilde{\pi}_j^{k,i}\right)}{\sum_{l=1}^{\mathbb{C}} \exp\left(\tilde{\pi}_l^{k,i}\right)} \quad (2)$$

**Intensity Posterior** $q_\phi(I^{k,i}|Z^{k,i}, X^{k,i})$  After sampling $I^{k,i} \sim q_\phi(I^{k,i}|Z^{k,i}, X^{k,i})$, the model applies a per-gesture, region-wise impact. This can be formulated in Eq. 3. where $H^{i-D^{k,i}/2:i+D^{k,i}/2,k}$ represents the local window of impact, $I^{k,i}$ is the sampled impact value, and $D^{k,i}$ is the duration of the gesture. We actually applied Sigmoid function to deliver positive intensity values. The Hann function is used to apply the impact smoothly within the local window. The motivation behind this formulation is that most patches $(k, i)$ are not activated, reflecting the sparse nature of human speech production and co-articulation [70, 77]. Visualizations can be checked in Appendix.A.2.2.

$$H^{i-\frac{D^{k,i}}{2}:i+\frac{D^{k,i}}{2},k} = \text{Hann}\left(\text{Sigmoid}(I^{k,i} \sim q_\phi(I^{k,i}|Z^{k,i}, X^{k,i})), D^{k,i} \sim q_\phi(D^{k,i}|Z^{k,i}, X))\right) \quad (3)$$

**Online Sparse Sampling**  Given the limited number of patches contributing to gestural scores [71], we localize the impact within a specific window. We define a *Combined Score* $S^{k,i} = aI^{k,i} + bD^{k,i}$, where $I^{k,i}$ and $D^{k,i}$ represent impact and duration, respectively, and $a$ and $b$ are hyperparameters. This score ranks the importance of each patch, with indices for each gesture computed as $r_{row}(k, i) = \text{rank}(-S^{k,i}$ within row $k)$. Setting $m_{row}$ as the number of patches selected, we apply a sparse mask $M_{row}$ (Eq. 4) to derive the final sparse gestural scores, detailed in Eq. 4. This entire online sparse sampling process is differentiable. The parameters $a$, $b$, and $m_{row}$ are elaborated in the Appendix. For simplicity, we denote this process as $(i, k) \sim \mathbb{S}$, with visualizations in Appendix A.2.3.

$$\tilde{H}^{k,i} = M_{row}^{k,i} \cdot H^{k,i} \qquad \text{where} \quad M_{row}^{k,i} = \begin{cases} 1 & \text{if } r_{row}(k, i) \leq m_{row} \\ 0 & \text{otherwise} \end{cases} \quad (4)$$

**Multi-scale Gestural Decoder**  The decoder reconstructs $\hat{X} = [\hat{X}_1, \hat{X}_2, ..., \hat{X}_t] \in \mathbb{R}^{d\times t}$ from gestures $G \in \mathbb{R}^{T\times d\times K}$ and gestural scores $\tilde{H} \in \mathbb{R}^{K\times t}$. In this work, we retain the CMF operation [77] and extend it to multiple deep layers. We also introduce multi-scale mechanism, which has proven to be a robust tokenizer for various speech tasks [92, 93, 62, 94]. Denote: $f_{\text{down},\theta}^{1/2}, f_{\text{down},\theta}^{1/4}, f_{\text{up},\theta}^{2}, f_{\text{up},\theta}^{4}$ as downsample/upsample modules with scales of $1/2$ or $1/4$. The convolutive matrix factorization operator $\mathcal{A} * \mathcal{B}$ means $\sum_{i=0}^{T-1} \mathcal{A}(i) \cdot \overrightarrow{\mathcal{B}}^i$ where $\mathcal{A} \in \mathbb{R}^{T\times d\times K}$ and activation function $\mathcal{B} \in \mathbb{R}^{K\times t}$. Then our multi-scale decoder is defined in Eq. 5, where $r = 1$ means no resolution change, and $f_{\text{trans}}$ represents any neural network, details of which can be found in the Appendix. Up to this point, $p_\theta(X|D, I, G)$ (Eq. 1) is defined. We provide more details in Appendix A.2.4.

$$\hat{X} = \sum_{r\in\{1,2,4\}} f_{\text{up},\theta}^r\left(f_{\text{trans},\theta}\left(G * f_{\text{down},\theta}^{1/r}(\tilde{H})\right)\right) \quad (5)$$

### 2.2.3  Gestural Scores as Phonetic Representations

After obtaining gestural scores, we predict phoneme alignment for dysfluency modeling. For clean speech, alignment is acquired using the Montreal Forced Aligner (MFA) [80], while for dysfluent

speech, it is simulated (see Section 5). The direct prediction of phoneme alignment from handcrafted features or self-supervised learning (SSL) units [51] is limited due to scalability issues with dysfluent speech, discussed further in Sec. 6. We utilize 4X downsampled gestural scores (from decoding), denoted as $\hat{H}$, matching the resolution of acoustic features [95]. Let $\tau = [\tau_1, \tau_2, \dots, \tau_{t'}]$ represent the phoneme alignment, where $t' = t/4$. Employing the Glow algorithm [96], we transform $\hat{H}$ into $\tau$, expressed as $\tau = f_\theta^G(\hat{H})$, optimized via a softmax crossentropy objective $\mathcal{L}_{\text{phn}}$.

**Self-Distillation**  We distill gestural scores from pretrained acoustic features [95], which are then adapted to match gestural scores' dimensions. Instead of directly measuring the distance between acoustic embeddings and gestural scores, we use the alignment-conditioned *gestural prior* as an acoustic-conditioned *gestural posterior*. The reference text $C = [C_1, C_2, \dots, C_L]$ is processed by a text encoder to yield the latent Gaussian posterior $(\mu_\theta^{C_1}, \sigma_\theta^{C_1}), (\mu_\theta^{C_2}, \sigma_\theta^{C_2}), \dots, (\mu_\theta^{C_L}, \sigma_\theta^{C_L})$, with the gestural posterior modeled via the change of variable property $f_\theta^G$ as described in Eq. 6. Intuition, detailed methodology and visualization can be view in Appendix. A.3.

$$p_\theta(\hat{H}|C) = p_\theta(\tau|C) \left| \det\left( \frac{\partial f_\theta^G(\hat{H})}{\partial \hat{H}} \right) \right| = \frac{1}{K_1} \prod_{i=1}^{t'} \prod_{j=1}^{L} \mathcal{N}\left( \tau_i; \mu_\theta^{C_j}, (\sigma_\theta^{C_j})^2 \right) \left| \det\left( \frac{\partial f_\theta^G(\hat{H})}{\partial \hat{H}} \right) \right| \quad (6)$$

Conversely, given the acoustic embedding $A = [A_1, A_2, \dots, A_L]$, a text encoder is employed to output the latent Gaussian posterior $(\mu_\theta^{A_1}, \sigma_\theta^{A_1}), (\mu_\theta^{A_2}, \sigma_\theta^{A_2}), \dots, (\mu_\theta^{A_{t'}}, \sigma_\theta^{A_{t'}})$. The posterior $q_\theta(\hat{H}|A)$ can be derived in a similar manner. The overall distillation loss is then presented in Eq. 7.

$$\mathcal{L}_{\text{dist}} = \text{KL}\left( q_\theta(\hat{H}|A) \| p_\theta(\hat{H}|C) \right), \quad \text{where} \quad q_\theta(\hat{H}|A) = \frac{1}{K_2} \prod_{i=1}^{t'} \prod_{j=1}^{t'} \mathcal{N}\left( \hat{H}_i; \mu_\theta^{A_j}, (\sigma_\theta^{A_j})^2 \right) \tag{7}$$

Both $K_1$ and $K_2$ are normalization terms. The overall loss objective for neural variational gestural modeling is shown in Eq. 8, where $\lambda_1, \lambda_2, \lambda_3$ are balancing factors.

$$\mathcal{L}_{\text{VAE}} = -\lambda_1 \mathcal{L}_{\text{ELBO}} + \lambda_2 \mathcal{L}_{\text{phn}} + \lambda_3 \mathcal{L}_{\text{dist}} \tag{8}$$

## 3 Connectionist Subsequence Aligner (CSA) for Dysfluency Modeling

### 3.1 Monotonic Alignment is effective Dysfluency Aligner

Given the reference text $C = [C_1, C_2, \dots, C_L]$ and dysfluent phonetic alignment $\tau = [\tau_1, \tau_2, \dots, \tau_{t'}]$, the alignment between $C$ and $\tau$ is typically non-monotonic. For example, when people say "pl-please," it is non-monotonically aligned with "p-l-e-a-s-e." Prior work [1, 82] on non-monotonic dysfluent modeling has its limitations, as discussed in Sec. 2.1. In this work, we focus on *monotonic alignment* and argue that it is effective dysfluency aligner. The intuition is straightforward: we seek an aligner $\gamma : \{1, 2, \dots, L\} \to \mathcal{P}(\{1, 2, \dots, t'\})$ such that for each $i \in \{1, 2, \dots, L\}$, Eq. 9 holds. The aligner $\gamma$ maps elements in $C$ to consecutive subsequences in $\tau$ without overlap. This property is beneficial for dysfluency detection, as for each element in $C$, we can determine the presence of dysfluencies such as insertion, deletion, repetition, block, replacement, etc., based on $\gamma(C_i)$.

$$\gamma(C_i) = [\tau_{s_i}, \tau_{s_i+1}, \dots, \tau_{e_i}] \quad \text{where} \begin{cases} 1 \leq s_i \leq e_i \leq t' & \\ e_i < s_{i+1} & \forall i \in \{1, 2, \dots, L-1\} \\ s_i < s_{i+1}, e_i < e_{i+1} & \forall i \in \{1, 2, \dots, L-1\} \end{cases} \tag{9}$$

### 3.2 Local Subsequence Alignment (LSA) Achieves Semantic Dysfluency Alignment

All monotonic aligners satisfy Eq.9, which serves as a necessary condition. However, we also desire $\gamma(C_i)$ to be semantically aligned with $C_i$. Consider the aforementioned example: one preferred alignment is $\gamma(p)=[p,l,p]$, indicating the presence of a stutter. In contrast, if $\gamma(p)=[p,l,p,l,e,a,s]$, it becomes challenging to identify any reasonable dysfluency, despite still satisfying Eq.9. In this work, we propose that *Local Subsequence Alignment (LSA)* is an effective approach for achieving semantically aligned $\gamma$. Before delving into the main topic, we propose and introduce two terms: (i) *Global Sequence Aligner (GSA)*, where the cost function involves the alignment of all elements in the sequence; this includes most sequence aligners such as DTW [97–99], CTC [79], and MFA [80]; and (ii) *Local Sequence Aligner (LSA)*, where the cost function involves only a subset of elements. One representative is longest common subsequence (LCS) alignment [100, 101].

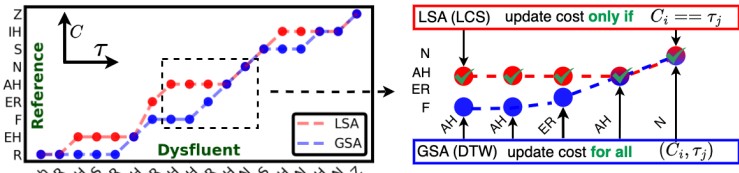

Figure 3: LSA(LCS) delivers dysfluent alignment that is more semantically aligned.

**Intuition** Fig. 3 (left) illustrates the effectiveness of LSA as a dysfluency aligner. The reference text $C$, a stress-free phoneme transcription [102] of word "references", contrasts with the dysfluent phonetic alignment $\tau$, which includes impairments like insertions of fillers and repetitions. LCS (LSA,[100]) and DTW (GSA,[97]) results are depicted in red and blue, respectively. LSA alignment $\gamma^{\text{LSA}}(C_i)$ shows higher semantic alignment with $C_i$ compared to DTW's $\gamma^{\text{GSA}}(C_i)$, which includes misaligned elements like an unwarranted alignment of "F". LSA's superiority stems from its cost function, which updates only for matching dysfluency-aware boundaries, while DTW updates for all pairs, often unrelated to dysfluency boundaries. Detailed analysis are available in Appendix A.7.

**Problem Statement** Taking LCS into our framework presents three challenges: *First*, the high dimensionality of $C$ and $\tau$ requires suitable emission and transition probability models. *Second*, LCS cost function is non-differentiable. *Third*, multiple LCS alignments necessitate effective modeling of joint distribution. To address these, we introduce *Connectionist Subsequence Aligner* (CSA).

### 3.3 Connectionist Subsequence Aligner (CSA) Formulation

**Objective** From gestural score $\hat{H}$, we obtain phonetic alignment $\tau = f_\theta^G(\hat{H}) = [\tau_1, \tau_2, \ldots, \tau_{t''}]$. In practice, both $\tau$ and $C$ are embeddings instead of explicit labels, where $C = [C_1, ..., C_L]$ are sampled from the text encoder $\mathcal{N}(\mu_\theta^{C_i}, (\sigma_\theta^{C_i})^2)$, $i = 1, ..., L$, as proposed in Sec.2.2.3. Let $t''$ denote the sequence length after removing duration from the original length $t'$. Duration will be reincorporated post-alignment. The alignment between $C$ and $\tau$ is already defined in Eq.9. We introduce another notation $\Gamma$, where $\Gamma(\tau_i)$ is the aligned token in $C$. $\Gamma(\tau) = [\Gamma(\tau_1), ..., \Gamma(\tau_{t''})]$ represents the final alignment with respect to $C$, in comparison to alignment $\gamma(C)$, which is with respect to $\tau$. There are possibly multiple ($N$) alignments $\gamma_j^{\text{LSA}}(C)$, where $j = 1, ..., N$. Our goal is

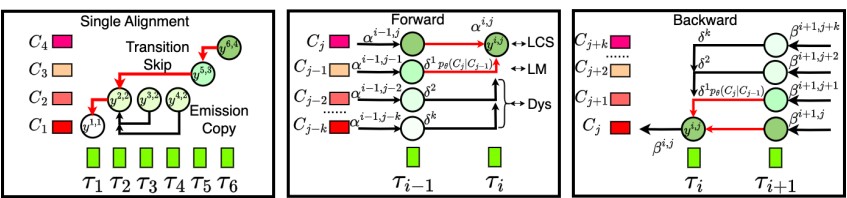

Figure 4: CSA

to optimize model $\theta$ to obtain the largest joint distribution of alignments $\sum_{j=1}^N \gamma_j^{\text{LSA}}(C)$. However, unlike CTC [79], we can't search alignments explicitly as the monotonic constraints are different. We propose approximating LSA. Let $\Gamma'(\tau)$ be one approximate LSA alignment, and assume there are $N$ possible LSA alignments: $\Gamma'_j(\tau)$ where $j = 1, ..., N$. Our final objective is formulated in Eq. 10.

$$\max_\theta \mathop{\mathbb{E}}_{C,\tau} \sum_{j=1}^N p_\theta(\gamma_j^{\text{LSA}}(C)|C,\tau) = \max_\theta \mathop{\mathbb{E}}_{C,\tau} \sum_{j=1}^N p_\theta(\Gamma_j^{\text{LSA}}(\tau)|C,\tau) \approx \max_\theta \mathop{\mathbb{E}}_{C\tau} \sum_{j=1}^N p_\theta(\Gamma'_j(\tau)|\tau) \quad (10)$$

**Approximate LSA Alignments** $\Gamma'(\tau)$ We define $y^{i,j}$ as the emission probability $p(C_j|\tau_i)$, and transition probability $p_\theta(\tau_j|\tau_i)$. Let $C_j^S$ denote the embedding sampled from the distribution $\mathcal{N}(\mu_\theta^{C_j}, (\sigma_\theta^{C_j})^2)$ (Sec.2.2.3). The emission probability is given in Eq. 11. We approximate the transition probability using a separate neural network $p_\theta(\tau_j|\tau_i)$.

$$y^{i,j} = p_\theta(C_j|\tau_i) \approx \frac{\exp^{\tau_i \cdot C_j^S}}{\left(\sum_{k=1}^L \exp^{\tau_i \cdot C_k^S}\right)} \quad (11)$$

It is possible to list all LCS alignments $\Gamma'_j(\tau)$, where $i = 1, ..., N$, via soft alignments [99], which are also differentiable. However, we propose that by simply introducing the LCS constraint on the vanilla CTC [79] objective, the LCS can be implicitly applied, which we call the *Connectionist Subsequence Aligner* (CSA). Let us consider Figure 4 (left) for intuition. For a single alignment $\Gamma'_j(\tau)$, the emission probability and transition probability will only be applied if $C_i$ is already aligned ($\tilde{C}_1$ in the figure). We refer to these as *Transition Skip* and *Emission Copy*. Now, let us move to the LCS-constrained forward-backward algorithm [79]. Taking the forward algorithm (Figure 4 (mid)) for illustration, *Emission Copy* is reflected in $\alpha_{i,j}$ via an identity multiplier on $\alpha_{i-1,j}$. *Transition Skip* is reflected on both $\alpha_{i-1,j}$ and $\alpha_{i-1,j-1}$, where we apply a transition on $\alpha_{i-1,j-1}$. This also implicitly leverages language modeling. We also consider all previous tokens $C_{j-2}, ..., C_{j-k}, ..., C_0$; however, no transition is applied, but a discounted factor $\delta^k$ is utilized instead. This indicates a significant jump (deletion), which we denote as a dysfluency module, although all other modules model dysfluencies equally. Forward and backward algorithms are displayed in Eq. 12 and Eq. 13.

$$\alpha_\theta^{i,j} = \alpha_\theta^{i-1,j} + \sum_{k=1}^{j} \delta^k \alpha_\theta^{i-1,j-k} \cdot y^{i,j} \cdot \left( p_\theta(C_{j-1}^S | C_j^S) \cdot \mathbf{1}_{\{k=1\}} + \mathbf{1}_{\{k \neq 1\}} \right) \tag{12}$$

$$\beta_\theta^{i,j} = \beta_\theta^{i+1,j} + \sum_{k=1}^{t'-j} \delta^k \beta_\theta^{i+1,j+k} \cdot y^{i,j} \cdot \left( p_\theta(C_j^S | C_{j+1}^S) \cdot \mathbf{1}_{\{k=1\}} + \mathbf{1}_{\{k \neq 1\}} \right) \tag{13}$$

We initialize $\alpha_{1,1} = \beta_{t'',L} = 1$, $\alpha(i,1) = 0 \quad \forall i > 0$, $\beta(i,1) = 0 \quad \forall i < t''$, and $\beta(1,j) = 0 \quad \forall j < L$. Our CSA objective is displayed in Eq. 14, where we take the summation over all reference tokens and time stamps.

$$\mathcal{L}_{\text{CSA}} = -\mathop{\mathbb{E}}_{C,\tau} \sum_{j=1}^{N} p_\theta(\Gamma'_j(\tau) | \tau) = -\sum_{i=1}^{t''} \sum_{j=1}^{L} \frac{\alpha_\theta^{i,j} \beta_\theta^{i,j}}{y^{i,j}} \tag{14}$$

**Sampling** As the alignment $\Gamma'(\tau)$ is required for the next module, it is necessary to sample it during training. Traditional beam search methods are impeded by reduced inference speeds. To mitigate this, we employ the Longest Common Subsequence (LCS) algorithm offline on $\tilde{C}^e$ and $\tau^e$ to derive the alignments. The final alignment is denoted as $\gamma(C_i^S) = [\tau_{s_i}^S, \ldots, \tau_{e_i}^S]$, as presented in Eq. 9. This methodology yields a sequence of inputs in the form of CSA-O = $[(C_1^S, \gamma(C_1^S)), \ldots, (C_L^S, \gamma(C_L^S))]$.

## 4 Language Models and Overall Training Objective

Following LTU [23], we utilize speech representations (alignment) $[(C_1^S, \gamma(C_1^S)), \ldots, (C_L^S, \gamma(C_L^S))]$ (Sec. 3.3), along with word-level timestamps, reference text $C$, and instruction $C^I$, as input to LLaMA-7B [103]. During the training process, we incorporate annotations that include per-word disfluency with timestamps. Our approach strictly adheres to the procedures outlined in [23] and employs Vicuna instruction tuning [104] with LoRA [105]. As this is not our core contribution, we provide details in Appendix A.8. We use the same autoregressive training objective as [23], denoted as $\mathcal{L}_{\text{LAN}}$. The overall loss objective for SSDM is shown in Eq. 15.

$$\mathcal{L}_{\text{SSDM}} = \mathcal{L}_{\text{VAE}} + \mathcal{L}_{\text{CSA}} + \mathcal{L}_{\text{LAN}} \tag{15}$$

## 5 Libri-Dys: Open Sourced Dysfluency Corpus

Traditional rule-based simulation methods [1, 37, 50] operate in acoustic space, and the generated samples are not naturalistic. We developed a new pipeline that simulates in text space. To achieve this, we first convert a sentence into an IPA phoneme sequence. Then, we develop TTS rules for phoneme editing to simulate dysfluency, providing five types of dysfluency: Repetition(phoneme & word), Missing(phoneme & word), Block, Replacement and Prolongation. These rules are applied to the entire LibriTTS dataset [106], allowing the voice of generated speech to vary from the 2456 speakers included in the LibriTTS. The TTS-rules, entire pipeline, dataset statistics, MOS evaluation and phoneme recognition results are available in Appendix A.9. Overall Libri-Dys is 7X larger than LibriTTS, with a total size of 3983 hours. Data is opensourced at `https://bit.ly/4aoLdWU`.

# 6 Experiments

## 6.1 Data Setup

For training, we use VCTK++[1] and Libri-Dys datasets. For testing, we randomly sample 10% of the training data. Additionally, we incorporate nfvPPA[107] data from our clinical collaborations, which includes 38 participants—significantly more than the 3 speakers in prior studies [1, 2]. It is approximately 1 hour of speech. Further details are provided in Appendix A.10.1.

## 6.2 Experiments Setup

The neural gestural VAE (Eq.8), CSA (Eq.14), and language modeling components are trained sequentially, with each stage completed before the next begins. Subsequently, we perform end-to-end learning to implement curriculum learning. Our objective is to evaluate the dysfluent intelligibility and scalability of our proposed *gestural scores*, as well as the dysfluency detection performance of each proposed module. We evaluate phonetic transcription and alignment using the framewise **F1 Score** and Duration-Aware Phoneme Error Rate (**dPER**). The F1 Score measures how many phonemes are correctly predicted, while dPER extends the traditional Phoneme Error Rate (PER) by assigning specific weights to different types of errors. For dysfluency evaluation, besides F1 Score, we also report the time-aware Matching Score (**MS**), which measures both type and temporal accuracy, with temporal matching considering the Intersection over Union (IoU) threshold of 0.5. Detailed training configurations can be found in Appendix A.12.

## 6.3 Scalable Intelligibility Evaluation

| Method | Eval Data | F1 (%, ↑) | dPER (%, ↓) | F1 (%, ↑) | dPER (%, ↓) | F1 (%, ↑) | dPER (%, ↓) | F1 (%, ↑) | dPER (%, ↓) | F1 (%, ↑) | dPER (%, ↓) | SF1 (%, ↑) | SF2 (%, ↓) |
|---|---|---|---|---|---|---|---|---|---|---|---|---|---|
| Training Data | | *VCTK++* | | *LibriTTS (100%)* | | *Libri-Dys (30%)* | | *Libri-Dys (60%)* | | *Libri-Dys (100%)* | | | |
| HuBERT [108] | VCTK++ | 90.5 | 40.3 | 90.0 | 40.0 | 89.8 | 41.2 | 91.0 | 40.2 | 89.9 | 41.2 | 0.15 | -0.1 |
| | Libri-Dys | 86.2 | 50.3 | 88.2 | 47.4 | 87.2 | 42.3 | 87.2 | 43.4 | 87.8 | 42.9 | 0.18 | 0.29 |
| H-UDM [2] | VCTK++ | 91.2 | 39.8 | 91.0 | 38.8 | 90.7 | 39.0 | 91.3 | 39.9 | 90.9 | 40.2 | 0.12 | 0.45 |
| | Libri-Dys | 88.1 | 44.5 | 88.9 | 45.6 | 88.0 | 43.3 | 88.5 | 43.3 | 88.9 | 43.0 | 0.32 | -0.09 |
| GS-only | VCTK++ | 88.1 | 41.9 | 88.1 | 42.2 | 88.3 | 41.9 | 88.9 | 41.9 | 89.4 | 40.7 | 0.39 | -0.36 |
| | Libri-Dys | 84.7 | 44.5 | 85.0 | 43.3 | 85.5 | 43.0 | 85.7 | 42.2 | 86.5 | 41.5 | 0.32 | -0.53 |
| GS w/o dist | VCTK++ | 91.4 | 39.0 | 91.6 | 38.5 | 91.5 | 38.8 | 92.0 | 37.2 | 92.6 | 37.1 | 0.38 | -0.67 |
| | Libri-Dys | 88.0 | 42.4 | 88.3 | 41.9 | 88.7 | 41.0 | 88.9 | 39.4 | 90.0 | 39.0 | 0.11 | **-0.76** |
| GS w/ dist | VCTK++ | **91.5** | **39.0** | **91.7** | **38.3** | **91.7** | **38.6** | **92.1** | **37.0** | **93.0** | **37.0** | 0.43 | -0.64 |
| | Libri-Dys | **88.2** | **40.9** | **88.9** | **40.9** | **89.0** | **40.8** | **89.2** | **39.0** | **90.8** | **39.0** | **0.56** | -0.72 |

Table 1: Scalable Dysfluent Phonetic Transcription Evaluation

We evaluate phonetic transcription (forced alignment) performance using simulated data from VCTK++[1] and our proposed Libri-Dys dataset. The framewise F1 score and dPER[1] are used as evaluation metrics. Five types of training data are used: VCTK++, LibriTTS (100%, [106]), Libri-Dys (30%), Libri-Dys (60%), and Libri-Dys (100%). HuBERT [108] SSL units and H-UDM alignment (WavLM [95]) fine-tuned with MFA [80] targets are adopted. Additionally, we examine Gestural Scores (GS). GS-only refers to gestural VAE training (Eq.1), GS w/o dist excludes $\mathcal{L}_{\text{dist}}$, and GS w/ dist includes it, following Eq.8. Results are presented in Table 1. H-UDM consistently outperforms HuBERT due to the WavLM backbone. Gestural scores from Eq. 1 show inferior results due to sparse sampling. However, GS demonstrates better scalability compared to SSL units. Using phoneme alignment loss $\mathcal{L}_{\text{phn}}$ significantly increases intelligibility, matching SSL unit results. GS outperforms SSL units with more training data. The inclusion of the self-distillation objective yields the best performance and scalability. Scaling factors SF1 for F1 score and SF2 for dPER are computed as $(c - b) \times 0.3 + (b - a) \times 0.4$ for results [a, b, c] from Libri-Dys [30%, 60%, 100%]. In terms of intelligibility, Gestural Score delivers the best scalability.

| Method | Eval Data | F1 (%, ↑) | MS (%, ↑) | F1 (%, ↑) | MS (%, ↑) | F1 (%, ↑) | MS (%, ↑) | F1 (%, ↑) | MS (%, ↑) | F1 (%, ↑) | MS (%, ↑) | SF1 (%, ↑) | SF2 (%, ↑) |
|---|---|---|---|---|---|---|---|---|---|---|---|---|---|
| Training Data | | *VCTK++* | | *LibriTTS (100%)* | | *Libri-Dys (30%)* | | *Libri-Dys (60%)* | | *Libri-Dys (100%)* | | | |
| H-UDM [2] | VCTK++ | 78.3 | 60.7 | 82.5 | 63.9 | 84.3 | 66.1 | 84.2 | 65.3 | 84.1 | 65.2 | -0.07 | -0.35 |
| | Libri-Dys | 74.8 | 63.9 | 75.0 | 62.9 | 77.2 | 60.1 | 75.0 | 62.3 | 75.9 | 61.1 | -0.61 | 0.64 |
| SSDM | VCTK++ | 84.8 | 64.3 | 87.8 | 68.2 | 88.5 | 69.7 | 89.0 | 69.9 | 89.2 | 70.2 | 0.26 | 0.17 |
| | Libri-Dys | 78.9 | 68.3 | 79.0 | 69.4 | 79.3 | 69.8 | 80.6 | 69.9 | 81.4 | 70.4 | 0.76 | 0.19 |
| w/o LLaMA | VCTK++ | 84.5 | 64.0 | 86.9 | 68.0 | 88.4 | 69.7 | 88.7 | 69.8 | 88.9 | 69.9 | 0.18 | 0.07 |
| | Libri-Dys | 78.2 | 68.1 | 78.3 | 69.0 | 78.8 | 69.2 | 79.6 | 69.3 | 80.7 | 70.0 | 0.65 | 0.25 |
| w/ DTW | VCTK++ | 80.3 | 60.9 | 83.5 | 65.9 | 84.2 | 66.2 | 85.0 | 66.6 | 85.2 | 67.2 | 0.38 | 0.34 |
| | Libri-Dys | 75.9 | 65.6 | 76.3 | 67.4 | 76.7 | 67.5 | 77.9 | 68.2 | 78.0 | 68.4 | 0.51 | 0.32 |
| w/o GS | VCTK++ | 84.3 | 64.1 | 86.9 | 65.0 | 87.4 | 66.2 | 87.1 | 66.3 | 87.2 | 66.5 | -0.09 | 0.1 |
| | Libri-Dys | 76.9 | 66.1 | 77.0 | 66.4 | 77.7 | 67.8 | 78.6 | 68.1 | 78.8 | 68.4 | 0.42 | 0.21 |
| w/ Curri | VCTK++ | **85.6** | **65.1** | **87.1** | **68.5** | **88.8** | **69.9** | **89.2** | **70.2** | **90.0** | **71.9** | 0.4 | **0.63** |
| | Libri-Dys | **79.2** | **68.4** | **79.4** | **69.5** | **79.4** | **69.9** | **81.0** | **70.5** | **81.6** | **71.0** | **0.82** | 0.39 |

Table 2: Scalable Dysfluent Detection Evaluation (Simulation)

## 6.4  Scalable Dysfluency Evaluation

We follow [2] by using F1 (type match) and MS (matching score). The matching score is defined as follows: if the IoU (Intersection over Union) between the predicted time boundary and the annotations is greater than or equal to 0.5, and the type also matches, it is considered detected. We use H-UDM [2], the current state-of-the-art time-aware dysfluency detection model, as the baseline. Under our SSDM framework, we include several ablations: (1) We remove LLaMA and use a template matching algorithm [2] on top of CSA alignments; (2) We replace CSA with softDTW [99]; (3) We replace gestural scores with WavLM [95] units; (4) We adopt curriculum training, first training the gestural VAE, CSA, and LLaMA separately, then training them end-to-end. For language model outputs, we set the prompt and use [109] to automatically extract both types and time information from the response.The results in Table 2 show similar trends in terms of both performance and scalability (SF1 and SF2). Notably, we observe that LLaMA modeling does not contribute significantly, while both gestural scores and CSA (especially the latter) contribute the most. t is also noted that dysfluent phonetic intelligibility, as shown in Table 1, is highly correlated with detection performance.

## 6.5  State-of-the-art Dysfluency Detection

We select the optimal configuration and compare it with state-of-the-art speech understanding systems. For fair comparison, we fine-tune LTU-AS-13B [24] and SALMONN-13B [27] using the same instructions but with pure speech input (AST [110] for LTU-AS and Whisper [17] for SALMONN). Additionally, we attach a time embedding to model temporal aspects. Detailed information is available in Appendix A.8. We also test on real nfvPPA speech, with results presented in Table 3. Current large-scale models [24, 27, 109] show limited performance in dysfluent speech detection, as shown in Fig. 1. The detection of nfvPPA speech remains challenging due to the significant gap between simulated and real disordered speech. See our demo at `https://berkeley-speech-group.github.io/SSDM/`.

| Eval Data | LTU-AS-13B [24] | | LTU-AS-13B-FT | | SALMONN-13B [27] | | SALMONN-13B-FT | | ChatGPT [109] | | SSDM | | SSDM w/ Curri | |
|---|---|---|---|---|---|---|---|---|---|---|---|---|---|---|
| | F1(%, ↑) | MS(%, ↑) | F1(%, ↑) | MS(%, ↑) | F1(%, ↑) | MS(%, ↑) | F1(%, ↑) | MS(%, ↑) | F1(%, ↑) | MS(%, ↑) | F1(%, ↑) | MS(%, ↑) | F1(%, ↑) | MS(%, ↑) |
| VCTK++ | 7.2 | 0 | 12.2 | 1.7 | 7.3 | 0 | 14.2 | 0.5 | 25.3 | 0 | 89.2 | 70.2 | **90.0** | **71.9** |
| Libri-Dys | 8.9 | 0 | 9.7 | 1.7 | 7.7 | 0 | 11.0 | 2.5 | 18.3 | 0 | 81.4 | 70.4 | **81.6** | **71.0** |
| nfvPPA | 0 | 0 | 2.4 | 0 | 0 | 0 | 1.8 | 0 | 5.6 | 0 | 69.2 | 54.2 | **69.9** | **55.0** |

Table 3: Detection results from state-of-the-art models.

## 6.6  Dysfluency Visualization

We attempt to visualize dysfluency in gestural space, as shown in Fig. 5. The correct text is "please" (p l i: z), while the real dysfluent speech is (p l e z). We apply GradCAM [111] to visualize the gradient of gestural scores $H$, shown in the right figure. We select the specific gestural scores corresponding to the vowel 'i' (e), and then

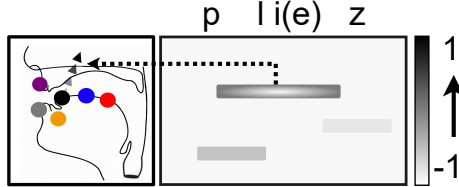

Figure 5: Gestural Dysfluency Visualization

visualize the corresponding gesture. On the gestural score, the gradient is negative in the center, indicating that the tongue is attempting to move down, which is the incorrect direction for articulation. This observation is meaningful as it provides insight into the dysfluency. Our system also offers explainability and has the potential to serve as a more interactive language learning tool.

## 7  Limitations and Conclusions

In this work, we proposed SSDM (Scalable Speech Dysfluency Modeling), which outperforms the current best speech understanding systems by a significant margin. However, there are still several limitations. First, we utilize LLMs, whose contribution is mariginal and whose potential has not been fully leveraged. We suspect this is due to the granularity of tokens, and we believe it would be beneficial to develop a phoneme-level language model to address this issue. Second, the current data scale is still inadequate, which is further constrained by computing resources. Third, we believe that learnable WFST [83, 82] could provide a more efficient and natural solution to this problem, yet it has not been extensively explored. Fourth, it is worthwhile to explore representations based on real-time Magnetic Resonance Imaging (rtMRI) [112] or gestural scores [78]. These approaches might enable the avoidance of the distillation process. Recent concurrent works have been focusing on region-based [113, 114] and token-based [115] approaches. It would be useful to explore the combination of these to leverage advantages on each side.

## Acknowledgments and Disclosure of Funding

Thanks for support from UC Noyce Initiative, Society of Hellman Fellows, NIH/NIDCD, and the Schwab Innovation fund.

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

# A    Appendix / supplemental material

## A.1    Gestural Modeling Visualization

### A.1.1    What are gestures and gestural scores?

The raw articulatory data $X \in \mathbb{R}^{12 \times t}$, where $t$ represents time with a sampling rate of 200 Hz, includes x and y coordinates of six articulators: Upper Lip, Lower Lip, Lower Incisor, Tongue Tip, Tongue Blade, and Tongue Dorsum. Here, $X$ is sourced from UAAI 2.2.1. As motion data, $X$ can be decomposed into gestures $G$ and gestural scores $H$. $T$ represents a window size that is much smaller than $t$, set at $T = 200$ ms. Fig. 6 provides an example with only three gestures: $G_1$ corresponds to a 200 ms trajectory of upper lip movement, $G_2$ to the lower lip, and $G_3$ to the tongue dorsum. This is illustrative; in our work, we use 40 gestures, approximately the size of the CMU phoneme dictionary [102] excluding stress. The gestural score $H \in \mathbb{R}^{3 \times t}$, where 3 corresponds to three gestures with corresponding indices. The first row represents the duration and intensity of gesture 1, the upper lip movement, and similarly for gestures 2 and 3. After decomposition, we obtain gestural scores as *representations*, which include duration and intensity, supporting co-articulation from different articulators with potential overlap. These scores are typically sparse and under certain conditions, they are also interpretable [77]. The next question is: where do these gestures come from? We performed k-means clustering on the original data $X$. Specifically, we stacked every 200 ms segment of data from $X$ into a supervector. Then, we applied k-means clustering to all these supervectors across the entire dataset $X$. A simple example to understand gestures and gestural scores might be this: if a simplified dictionary includes the gestures "lower lower lip" and "raise upper lip," each with a duration of 1 second and normalized intensity, these would occur simultaneously for that duration.

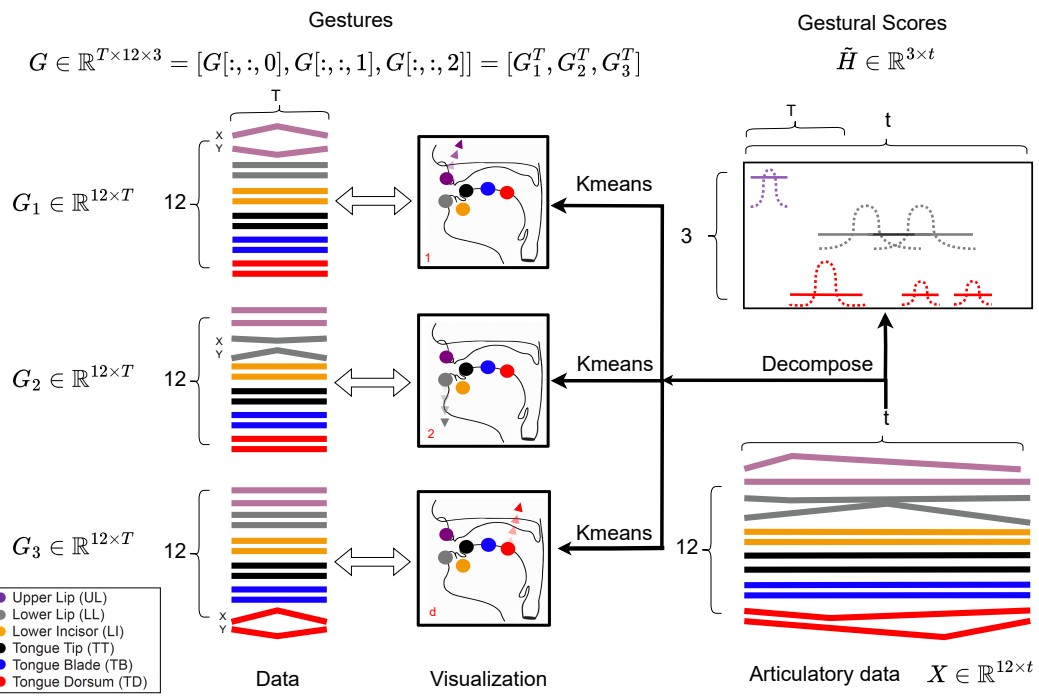

Figure 6: Gestures, Gestural Scores, Raw Data Visualization

### A.1.2    How does decomposition work?

As shown in Fig.7, convolutive matrix factorization (CMF)[75] decomposes $X \in \mathbb{R}^{12 \times t}$ into gestures $G \in \mathbb{R}^{T \times 12 \times 3}$ and gestural scores $H \in \mathbb{R}^{3 \times t}$. The original CMF algorithm[75] iteratively updates $G$ and $H$. For illustrative purposes, let us consider the reverse process. Given $G$, we select $G[:, i, :] \in$

$\mathbb{R}^{3 \times T}$ for $i = 1, 2, ..., 12$. Taking $G[:, 0, :]$ as an example, then $X[0][k] = G[:, 0, :] * \tilde{H}[:, k : k + T]$, where $*$ denotes the element-wise product sum.

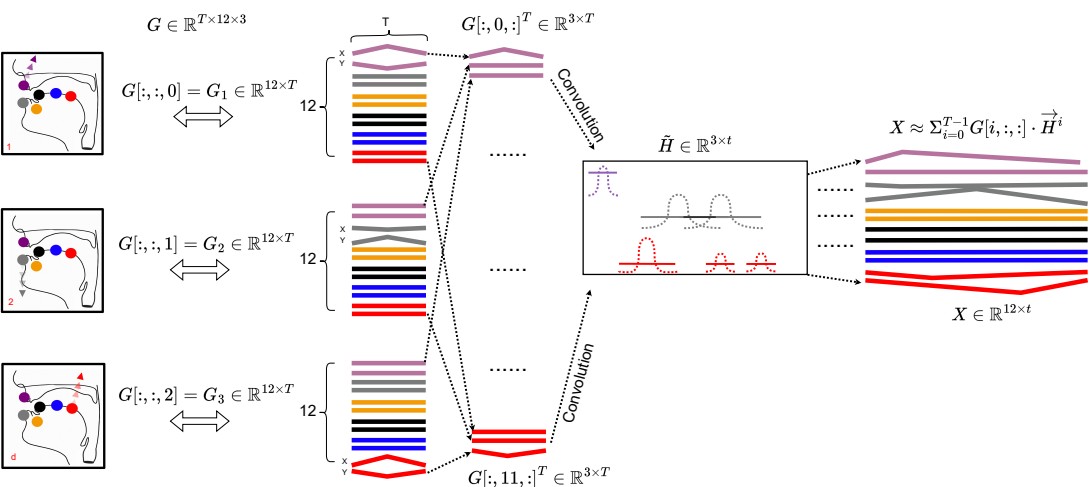

Figure 7: Illustration of Convolutive Matrix factorization

## A.2 Neural Variationl Gestural Modeling

[77] proposed using a neural network to predict $\tilde{H}$ and replacing the traditional CMF process with a one-layer 1-D convolutional neural network. However, strictly adhering to CMF results in a simplistic neural network architecture that limits the expressiveness of the learned gestural scores. We employ an advanced probabilistic encoder to predict $\tilde{H}$, implicitly modeling duration and intensity. Subsequently, we use a multi-scale decoder to simulate the CMF process and recover $X$. We will discuss the details in the following sections.

### A.2.1 Universal Acoustic to Articulatory Inversion (UAAI)

Since the real articulctory data are typically unavailable, we employ a state-of-the-art acoustic-to-articulatory inversion (AAI) model [88] pretrained on MNGU0 [84]. The model takes 16kHz raw waveform input and predicts 50Hz EMA features. We then upsample them to 200Hz. Although the AAI model was pretrained on *single speaker EMA*, it is a *universal template* for all speakers [88]. A concurrent study [116] further demonstrates that by performing speech-to-EMA-to-speech resynthesis, the single-speaker EMA representations derived from multi-speaker speech corpora, such as LibriTTS [106] and VCTK [117], maintain a sufficient level of intelligibility. Note that the entire system should be considered a *speech-only* system, as it does not require the use of any real EMA data during its operation.

### A.2.2 Implicit Duration and Intensity Modeling

However, there are three major differences between gestural scores and TTS duration modeling. First, unlike TTS, which enforces a monotonic alignment where each phoneme has a single duration, gestural scores permit independent gestures with overlapping durations, capturing co-articulation complexities [71]. Second, while TTS aligns text with speech, gestural scores lack a reference target for each gesture's duration. Third, durations in gestural scores are linked with intensities. Therefore, traditional differential duration modeling methods such as regression-based approaches [118–121], Gaussian upsampling [122–125], and variational dequantization [126–128] lead to unstable training in our setup.

Here we visualize our duration and intensity prediction modules in Fig. 8. Given the input $X \in \mathbb{R}^{12 \times t}$, a latent encoder is utilized to derive latent representations $Z \in \mathbb{R}^{(12 \times P) \times t}$. Subsequently, $Z$

is reshaped into a three-dimensional representation $Z \in \mathbb{R}^{12 \times P \times t}$, where $P$ denotes the patch embedding size for each patch index $(k, i)$, with $k = 1, \ldots, K$ representing the gesture index (in our configuration, $K = 40$), and $i$ is the time index. Each patch embedding is concatenated with $X[k, i]$, which is a scalar, to form a $P + 1$ embedding. This composite is then processed through the Intensity Encoder and Duration Encoder to predict their respective posteriors. For the intensity posterior, values are treated as floats. As gestural scores must always remain positive for enhanced interpretability, a Sigmoid function is applied to the sampled intensity $I^{k,i}$. The duration predictor operates as a classifier, where we establish the class set as $[1, 2, 3, \ldots, 50]$, thus constituting a 50-way classification problem. Due to the non-differentiable nature of the sampling process, we employ Gumbel Softmax [91] to generate the duration posterior. Consequently, for each point $(k, i)$ in the gestural score, we obtain a continuous positive intensity Sigmoid($I^{k,i}$) and a discrete duration $D^{k,i}$. A Hanning window is applied across the entire window: $H^{i - \frac{D^{k,i}}{2} : i + \frac{D^{k,i}}{2}, k} =$ Hann $\left( \text{Sigmoid}(I^{k,i} \sim q_\phi(I^{k,i}|Z^{k,i}, X^{k,i})), D^{k,i} \sim q_\phi(D^{k,i}|Z^{k,i}, X) \right)$. The Hanning window is defined as $w(n) = \text{Sigmoid}(I^{k,i}) \left( 1 - \cos\left( \frac{2\pi n}{D^{k,i} - 1} \right) \right)$, making it differentiable with respect to both $I^{k,i}$ and $D^{k,i}$.

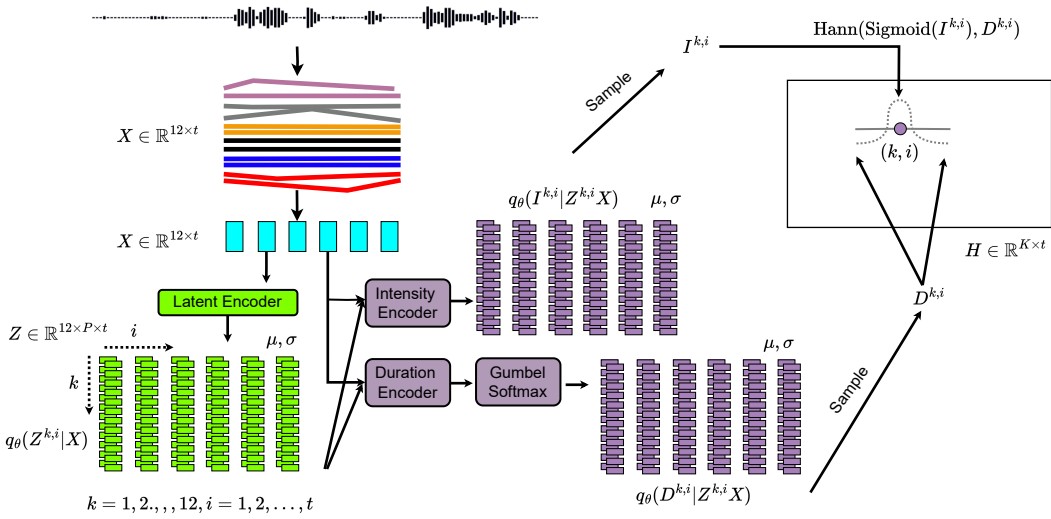

Figure 8: Duration and Intensity Modeling

### A.2.3 Sparse Sampling

The raw gestural scores $H$, as shown in Fig. A.2.2, represent a dense matrix. For each position $(k, i)$, a Hann window is applied, reflecting the inherent sparsity observed in human speech articulation [71]. To enhance this sparsity, we implement a sparse sampling method. As illustrated in Fig. 9, we define a ranking score $S^{k,i} = aI^{k,i} + bD^{k,i}$ where the coefficients are set to $a = 10$ and $b = 1$. We then select the top $m_{row}$ points based on the highest ranking scores. From this, we derive a Mask matrix $M_{row}$, which is applied to the original $H$ to produce a sparse gestural score $\tilde{H} \in \mathbb{R}^{K \times t}$.

### A.2.4 Multi-Scale Gestual Decoder

Traditional neural convolutional matrix factorization (CMF) [77] uses a single-layer neural network, which significantly limits the expressiveness of gestural modeling.

In this new decoder, we consider two sampling factors, $r = 2$ and $r = 4$. According to Eq. 5, Fig. 10 fully visualizes the multi-scale gestural decoder architecture. Note that the final representation $\hat{H}$ is downsampled by a factor of 4 to ensure consistency with the acoustic features from WavLM [95].

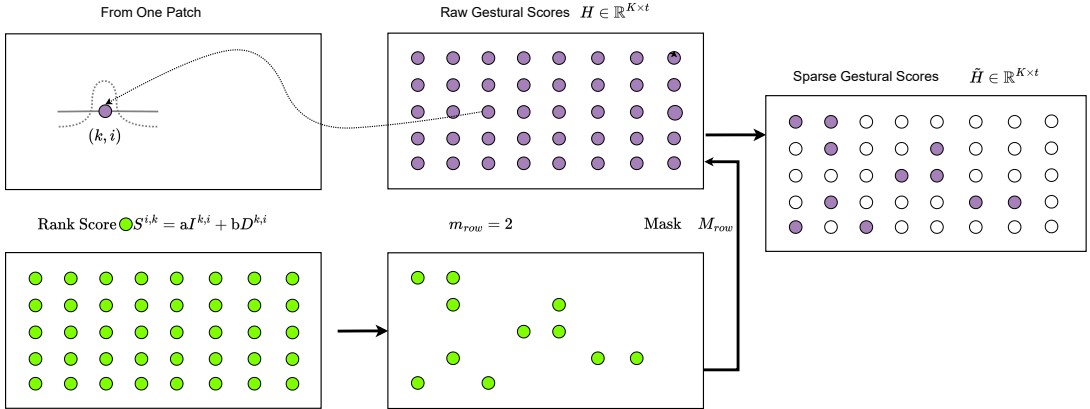

Figure 9: Sparse Sampling

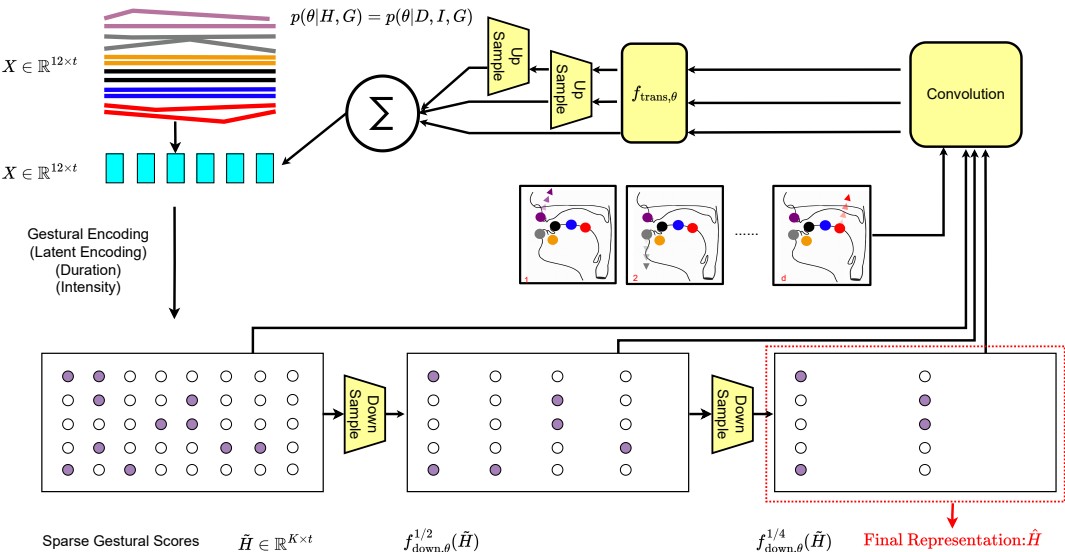

Figure 10: Multi-Scale Gestural Decoder

## A.3 Self-Distillation

**Background and Intuition** Electromagnetic articulography (EMA) data, sourced from either real samples [129] or UAAI [88], are typically sparsely sampled, leading to information loss. While concurrent work [116] achieves satisfactory intelligibility mostly at the word level, our objective is to enhance phonetic understanding. Furthermore, gestural scores precisely delineate phoneme boundaries [77]. This prompts the exploration of additional constraints to synergize these aspects. Self-distillation has been successful in computer vision [130–132] and speech [133–139], revealing *emergent properties* like unsupervised image segmentation [132] and speech semantic segmentation [139].

**Methods** We perform self-distillation per frame, indicated by the time index $i$ in Fig. 11. From the acoustic adaptor, we introduce the gestural score $\hat{H}$, which is downsampled by a factor of four, aligning it with the same resolution as speech features [95]. This allows us to derive the posterior $q_\theta(\hat{H}[i]|A[i])$. From the text encoder, which processes phonemes, we input the predicted phoneme embedding $\tau$ into the textual distribution parameters $\mu_\theta^C, \sigma_\theta^C$, obtaining $p_\theta(\tau_i|C)$ for each time step $i$.

Subsequently, through a flow and change of variable, we derive the prior distribution $p_\theta(\hat{H}|C)$. KL divergence is then applied between $p_\theta(\hat{H}[i]|C)$ and $q_\theta(\hat{H}[i]|A[i])$ to facilitate self-distillation.

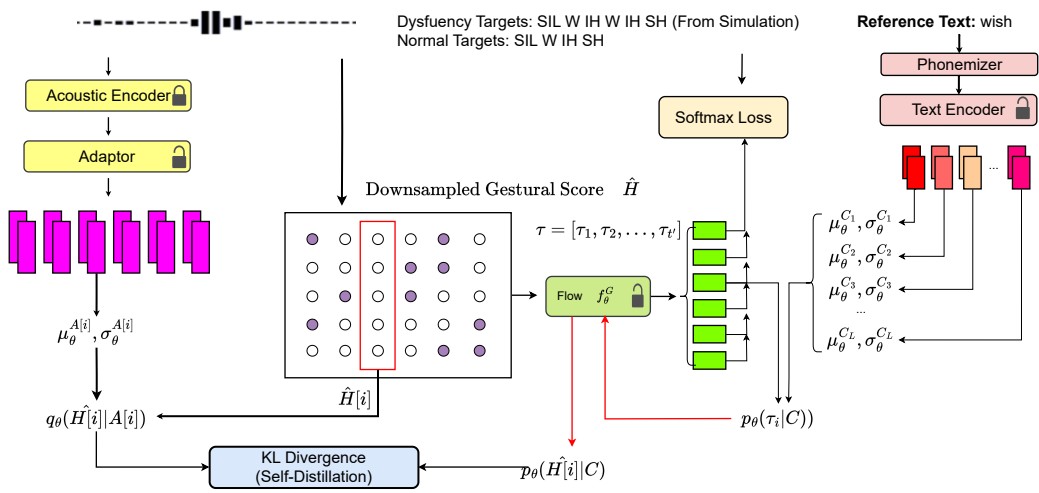

Figure 11: Self-Distillation Paradigm

## A.4 ELBO with latent variables

$$\log p_\theta(X) = \log \int p_\theta(X, Z, D, I)\, dZ\, dD\, dI \tag{16}$$

$$= \log \int q_\phi(Z, D, I|X) \frac{p_\theta(X, Z, D, I)}{q_\phi(Z, D, I|X)}\, dZ\, dD\, dI \tag{17}$$

$$= \log \mathbb{E}_{q_\phi(Z, D, I|X)} \left[ \frac{p_\theta(X, Z, D, I)}{q_\phi(Z, D, I|X)} \right] \tag{18}$$

$$\geq \mathbb{E}_{q_\phi(Z, D, I|X)} \left[ \log \frac{p_\theta(X, Z, D, I)}{q_\phi(Z, D, I|X)} \right] \quad \text{(Jensen's inequality)} \tag{19}$$

$$= \mathbb{E}_{q_\phi(Z, D, I|X)} \left[ \log p_\theta(X|D, I, G) \right] + \mathbb{E}_{q_\phi(Z, D, I|X)} \left[ \log \frac{p_\theta(Z, D, I)}{q_\phi(Z, D, I|X)} \right] \tag{20}$$

$$= \mathbb{E}_{q_\phi(Z, D, I|X)} \left[ \log p_\theta(X|D, I, G) \right] \tag{21}$$

$$\quad - \mathbb{E}_{(k,i)\sim\mathbb{S}} \left[ \mathrm{KL}\left( q_\phi(Z^{k,i}, D^{k,i}, I^{k,i}|X) \| p(Z^{k,i}, D^{k,i}, I^{k,i}) \right) \right] \tag{22}$$

$$= \mathbb{E}_{q_\phi(Z, D, I|X)} \left[ \log p_\theta(X|D, I, G) \right] \tag{23}$$

$$\quad - \mathbb{E}_{(k,i)\sim\mathbb{S}} \left[ \mathrm{KL}\left( q_\phi(Z^{k,i}|X) \| p(Z^{k,i}) \right) \right] \tag{24}$$

$$\quad - \mathbb{E}_{(k,i)\sim\mathbb{S}} \left[ \mathrm{KL}\left( q_\phi(D^{k,i}|X, Z^{k,i}) \| p(D^{k,i}) \right) \right] \tag{25}$$

$$\quad - \mathbb{E}_{(k,i)\sim\mathbb{S}} \left[ \mathrm{KL}\left( q_\phi(I^{k,i}|X, Z^{k,i}) \| p(I^{k,i}) \right) \right] \tag{26}$$

$$= \mathcal{L}_{\mathrm{ELBO}} \tag{27}$$

### A.5 LCS Pseudo Code

---

**Algorithm 1** Find Longest Common Subsequence (LCS)

---

**Require:** Target sequence `seq1`, Source sequence `seq2`
**Ensure:** Longest Common Subsequence (LCS) alignment
 1: Initialize a 2D array `lengths` of size $(\text{len(seq1)} + 1) \times (\text{len(seq2)} + 1)$ with zeros
 2: **for** each element $i$ in `seq1` **do**
 3:    **for** each element $j$ in `seq2` **do**
 4:       **if** `seq1[i] == seq2[j]` **then**
 5:          `lengths[i+1][j+1] = lengths[i][j] + 1`
 6:       **else**
 7:          `lengths[i+1][j+1] = max(lengths[i+1][j], lengths[i][j+1])`
 8:       **end if**
 9:    **end for**
10: **end for**
11: Initialize an empty list `align_lcs_result` to store the LCS alignment
12: Set $x = \text{len(seq1)}$, $y = \text{len(seq2)}$
13: **while** $x \neq 0$ **and** $y \neq 0$ **do**
14:    **if** `lengths[x][y] == lengths[x-1][y]` **then**
15:       $x = x - 1$
16:    **else if** `lengths[x][y] == lengths[x][y-1]` **then**
17:       `align_lcs_result.append((seq2[y-1], seq1[x-1]))`
18:       $y = y - 1$
19:    **else**
20:       `align_lcs_result.append((seq2[y-1], seq1[x-1]))`
21:       $x = x - 1$
22:       $y = y - 1$
23:    **end if**
24:    **if** $x == 0$ **and** $y \neq 0$ **then**
25:       `align_lcs_result.append((seq2[y-1], seq1[0]))`
26:       **break**
27:    **end if**
28:    **if** $x \neq 0$ **and** $y == 0$ **then**
29:       `align_lcs_result.append((seq2[0], seq1[x-1]))`
30:       **break**
31:    **end if**
32: **end while**
33: `align_lcs_result.reverse()`
34: **return** `align_lcs_result`

---

### A.6 DTW Pseudo Code

---

**Algorithm 2** Dynamic Time Warping (DTW)

---

**Require:** Sequence seq1, Sequence seq2)
**Ensure:** Alignment of seq1 and seq2
 1: $n \leftarrow$ len(seq1), $m \leftarrow$ len(seq2)
 2: Initialize dtw_matrix of size $(n + 1) \times (m + 1)$ with $\infty$
 3: dtw_matrix[0][0] = 0
 4: **for** $i = 1$ to $n$ **do**
 5:    **for** $j = 1$ to $m$ **do**
 6:       $cost \leftarrow$ distance(seq1[i-1], seq2[j-1])
 7:       dtw_matrix[i][j] $\leftarrow$ $cost +$ min(dtw_matrix[i-1][j], dtw_matrix[i][j-1], dtw_matrix[i-1][j-1])
 8:    **end for**
 9: **end for**
10: Initialize an empty list alignment to store the alignment
11: $i \leftarrow n, j \leftarrow m$
12: **while** $i > 0$ **and** $j > 0$ **do**
13:    **if** seq1[i-1] == seq2[j-1] **then**
14:       alignment.append((seq2[j-1], seq1[i-1]))
15:       $i \leftarrow i - 1$
16:       $j \leftarrow j - 1$
17:    **else**
18:       **if** dtw_matrix[i-1][j] == min(dtw_matrix[i-1][j], dtw_matrix[i][j-1], dtw_matrix[i-1][j-1]) **then**
19:          $i \leftarrow i - 1$
20:       **else if** dtw_matrix[i][j-1] == min(dtw_matrix[i-1][j], dtw_matrix[i][j-1], dtw_matrix[i-1][j-1]) **then**
21:          alignment.append((seq2[j-1], seq1[i-1]))
22:          $j \leftarrow j - 1$
23:       **else**
24:          alignment.append((seq2[j-1], seq1[i-1]))
25:          $i \leftarrow i - 1$
26:          $j \leftarrow j - 1$
27:       **end if**
28:    **end if**
29:    **if** $i == 0$ **and** $j > 0$ **then**
30:       **while** $j > 0$ **do**
31:          alignment.append((seq2[j-1], seq1[0]))
32:          $j \leftarrow j - 1$
33:       **end while**
34:       **break**
35:    **end if**
36:    **if** $i > 0$ **and** $j == 0$ **then**
37:       alignment.append((seq2[0], seq1[0]))
38:       **break**
39:    **end if**
40: **end while**
41: alignment.reverse()
42: **return** alignment

---

### A.7 Why LSA is better?

Fig. 3 provides an example and illustrates the intuition behind why LSA is a better candidate for dysfluency modeling. The reference text is the phoneme transcription of the word "references". To better illustrate this, the chosen dysfluent speech is significantly impaired, containing insertion of filler sounds such as "uh", repetition of sounds like "R", "EH", "AH", "IH", and insertion of sounds such as "S" and "ER", along with deletion of sounds like "F".

Let us examine the results from LSA and GSR. We aim to obtain all elements (phonemes) in the dysfluent alignment $\tau$ for each phoneme in the reference text $C$. LSA captures most dysfluencies through such per-reference alignment. For instance, the alignment (uh, R) to R indicates insertion. Similarly, (EH, S, R, EH) aligning to EH primarily indicates repetition. Up to this point, GSA exhibits similar performance, aligning (uh, R, EH, S, R) to R, which also indicates repetition. However, a significant difference emerges thereafter. For the phoneme F in the reference, no phonemes are aligned in LSA, which is correct as it is missing. Conversely, GSA aligns (ER, AH, AH) to F, which is unreasonable. For the phoneme AH in the reference, the LSA alignment (AH, AH, ER, AH) indicates repetition, which GSA fails to capture. Similarly, the repetition of IH is accurately captured by LSA but is missing in GSA. Our main point is that, although dysfluency alignment with the reference text is non-monotonic, aligning corresponding phonemes with each phoneme in the reference monotically enables fine-grained dysfluency analysis, which is naturally captured by LSA. Note that in Fig. 3, we use LCS [100] and DTW [97] for illustration.

Also look at Fig. 3 (right), we select a subsequence $C = [C_1, C_2, C_3, C_4]$ =[ER, AH, N, S] and $\tau = [\tau_1, \tau_2, \tau_3, \tau_4, \tau_5]$ =[AH, AH, ER, AH, N] from Fig. 3 (left), and provide an illustrative breakdown in Fig. 3 (right). LCS updates the cost function only when $C_3 = \tau_1 =$ AH and $C_4 = \tau_5 =$ N, excluding the remaining phonemes $[\tau_2, \tau_3, \tau_4] = [\text{AH}, \text{ER}, \text{AH}]$ from the cost function, as they are not essential for determining the alignment boundary. This is particularly relevant for $\tau_3 =$ ER, which is unrelated to the reference phoneme $C_3 =$ AH. In contrast, DTW considers all phonemes $\tau = [\tau_1, \tau_2, \tau_3, \tau_4] =$ [AH, AH, ER, AH] equally. While $\tau_2 =$ AH and $\tau_4 =$ AH are not crucial for deciding the boundary, their inclusion leads to a lower cost and higher weight in contributing to the final alignment. Therefore, LSA's selective cost function updates prove more effective for dysfluency alignment compared to DTW's equal consideration of all phonemes. Pseudo code is provided in Appendix. 1 and Appendix. 2 respectively.

### A.8 Language Models

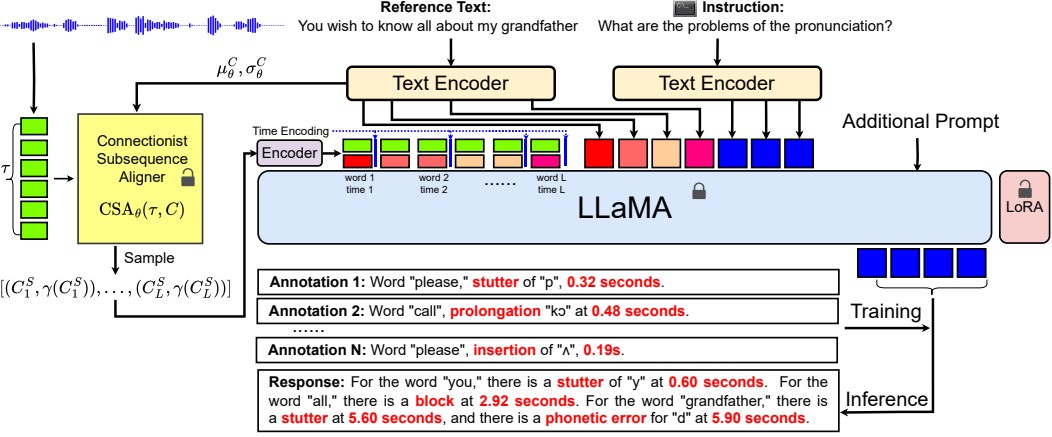

Figure 12: Instruction Tuning

### A.9 Dysfluency Simulation

#### A.9.1 TTS-rules

We inject dysfluency in the text space following these rules:

- **Repetition** (phoneme&word): The first phoneme or syllable of a randomly picked word was repeated 2-4 times, with pauselengths varying between 0.5 to 2.0 seconds.

- **Missing** (phoneme&word): We simulated two phonological processes that characterize disordered speech[140] - weak syllable deletion (deletion of a random unstressed syllable based on stress markers[1]) and final consonant deletion.

- **Block**: A duration of silence between 0.5-2.0 seconds was inserted after a randomly chosen word in between the sentence.

- **Replacement** (phoneme): We simulated fronting, stopping, gliding, deaffrication - processes that characterize disordered speech [141] - by replacing a random phoneme with one that would mimic the phonological processes mentioned above.

- **Prolongation** (phoneme): The duration of a randomly selected phoneme in the utterance was extended by a factor randomly chosen between 10 to 15 times its original length, as determined by the duration model.

#### A.9.2 Simulation pipeline

The simulation pipelines can be divided into following steps: **(i) Dysfluency injection:** We first convert ground truth reference text of LibriTTS into IPA sequences via the phonemizer [2], then add different types of dysfluencies at the phoneme level according to the *TTS rules*. **(ii) StyleTTS2 inference:** We take dysfluency-injected IPA sequences as inputs, conduct the StyleTTS2 [125] inference procedure and obtain the dysfluent speech. **(iii) Annotation:** We retrieve phoneme alignments from StyleTTS2 duration model, annotate the type of dysfluency on the dysfluent region. We show two samples (waveform and corresponding annotation) on the right side of the figure above.

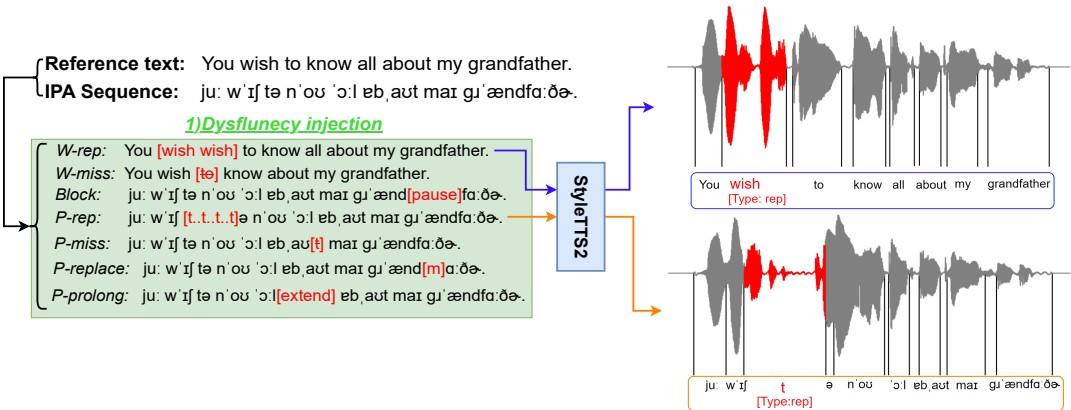

Figure 13: Simulation Pipeline

#### A.9.3 Datasets Statistics

The specific statistics of Libri-Dys are listed in Table. 4, compared with VCTK++. Figure. 14 presents a comparison between our simulated dataset and two existing simulated dysfluency datasets: VCTK++ [1] and LibriStutter [37]. It indicates that our dataset surpasses the datasets in both hours and the types of simulated dysfluencies. Note that since we build dataset based on publicly available corpus LibriTTS [106] and styletts2 [125], it satisfies safeguards criterion.

---

[1]https://github.com/timmahrt/pysle

[2]https://pypi.org/project/phonemizer/

Table 4: Types of Dysfluency Data in VCTK++ and Libri-Dys

| Dysfluency | # Samples VCTK++ [1] | Percentage VCTK++ | # Samples Libri-Dys | Percentage Libri-Dys |
|---|---|---|---|---|
| Prolongation | 43738 | 33.28 | 288795 | 13.24 |
| Block | 43959 | 33.45 | 345853 | 15.97 |
| Replacement | 0 | 0 | 295082 | 13.63 |
| Repetition (Phoneme) | 43738 | 33.28 | 340916 | 15.75 |
| Repetition (Word) | 0 | 0 | 301834 | 13.94 |
| Missing (Phoneme) | 0 | 0 | 296076 | 13.68 |
| Missing (Word) | 0 | 0 | 296303 | 13.69 |
| Total Hours of Audio | 130.66 | | 3983.44 | |

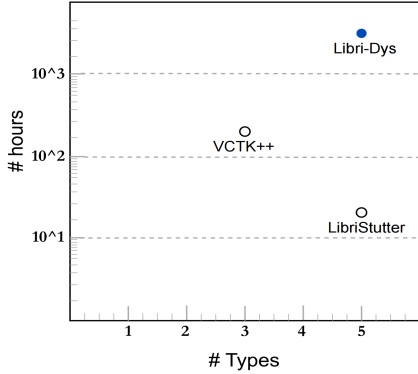

Figure 14: Existing Simulated dysfluency datasets

### A.9.4 Evaluation

To evaluate the rationality and naturalness of Libri-Dysefluency and use VCTK++ for comparison, we collected Mean Opinion Score (MOS, 1-5) ratings from 12 people. The final results are as displayed in Table. 5. Libri-Dys was perceived to be far more natural than VCTK++ (MOS of 4.15 compared to 2.14).

Table 5: MOS for VCTK++ [1] & Libri-Dys Samples

| Dysfluency Type | VCTK++ MOS | Libri-Dys MOS |
|---|---|---|
| Block | 2.66 ± 0.94 | 3.20 ± 1.26 |
| Missing (phoneme) | N/A | 4.66 ± 1.06 |
| Missing (word) | N/A | 4.80 ± 0.63 |
| Prolong | 1.33 ± 0.47 | 3.83 ± 0.89 |
| Repetition (phoneme) | 1.33 ± 0.43 | 4.33 ± 0.94 |
| Repetition (word) | N/A | 3.73 ± 1.49 |
| Replacement | N/A | 3.90 ± 0.99 |
| Overall | 2.14 ± 0.64 | 4.15 ± 0.93 |

### A.9.5 Phoneme Recognition

In order to verify the intelligibility of Libri-Dys, we use phoneme recognition model [142] to evaluate the original LibriTTS test-clean subset and various types of dysfluent speech from Libri-Dys. The Phoneme Error Rate (PER) is calculated and presented in Table. 6.

Table 6: Phoneme Transcription Evaluation on LibriTTS and Libri-Dys

| | LibriTTS | Libri-Dys | | | | |
|---|---|---|---|---|---|---|
| Type | / | W/P-Repetition | W/P-Missing | Block | Prolongation | Replacement |
| PER (% ↓) | 6.106 | 6.365 / 11.374 | 8.663 / 6.537 | 12.874 | 6.226 | 8.001 |

## A.10 Experiments

### A.10.1 nfvPPA

In looking for clinical populations to test our pipeline, we decided to focus on patients with a neurodegenerative disease called nonfluent variant primary progressive aphasia (nfvPPA). This phenotype falls under the umbrella of primary progressive aphasia (PPA), which is a neurodegenerative disease characterized by initially having most prominent disturbances to speech and language functions. PPA has three distinctive variants that correspond with unique clinical characteristics and differential patterns of brain atrophy: semantic (svPPA), logopenic (lvPPA), and nonfluent (nfvPPA) [107]. Disturbances to speech fluency can occur due to multiple underlying causes subsuming different speech and language subsystems in all of these variants; however, the variant most commonly associated with dysfluent speech is nfvPPA. This phenotype is characterized by primary deficits in syntax, motor speech (i.e., in this case, apraxia of speech), or both, and it is this association with apraxia of speech that makes nfvPPA an apt clinical target for assessing automatic processing of dysfluent speech.

Our collaborators regularly recruit patients with this disease as a part of an observational research study where participants undergo extensive speech and language testing with a qualified speech-language pathologist. This testing includes a comprehensive motor speech evaluation, which includes an oral mechanism exam, diadochokinetic rates, maximum phonation time, multisyllabic word reading, word of increasing length, passage reading, and connected speech samples. For our present purposes, we have decided to analyze the speech of participants reading aloud the Grandfather Passage, a passage often used clinically to assess motor speech due to its inclusion of nearly all phonemes of the English language. We have recordings for 10 participants with nfvPPA under IRB with consents signed for educational use. Passage recordings are conducted using high quality microphones for both in-person and remote visits. We randomly select 10 recordings and calculated the occurrences of various dysfluency types within them. The distribution is shown in Fig. 15. Note that nfvPPA data will not be released.

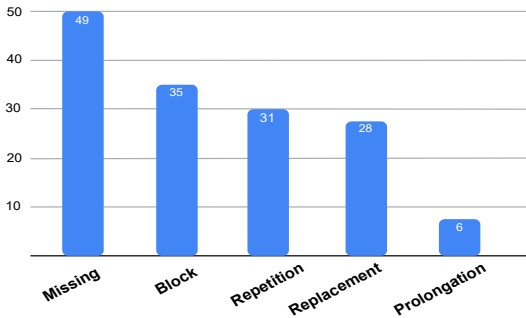

Figure 15: Dysfluency distribution in nfvPPA

## A.11 Model Configurations

The EMA features are denoted as $X = [X_1, X_2, ..., X_t]$, where $X_i \in \mathbb{R}^d$. These features represent the positions of 6 articulators, including the upper lip, lower lip, lower incisor, tongue body, tongue tip, and tongue dorsum, with both x and y coordinates. Consequently, $d$ is equal to 12.

### A.11.1 Acoustic Adaptor

We use WavLM [26] large as a pretrained acoustic encoder. The acoustic adaptor is a simple linear layer with dimensions (784, 40), where 40 is the number of gestures used in this paper. The output of the acoustic adaptor is $H \in \mathbb{R}^{40 \times t}$.

### A.11.2 Gestural Encoder

The latent encoder $q_{Z|X}$ is a 4-layer transformer with an input size of 12, hidden size of 96, and output dimension of 144. The latent representation is $Z \in \mathbb{R}^{12 \times 12 \times t}$, where the patch size is $P$. Sinusoidal positional encoding [143] is added to the input of each transformer layer to provide position information. The intensity encoder is a three-layer MLP with input dimension 12, hidden dimensions [24, 48], and outputs a scalar. The duration predictor is a 3-layer transformer with input dimension 12, hidden size 48, and outputs a 50-class distribution (0-49 duration bins). Sinusoidal positional encoding is added to the input of each transformer layer.

### A.11.3 Gestural Decoder

Downsampling is performed by average pooling (every 2 or 4 frames). Upsampling is performed using a deconvolutional layer (also known as transposed convolution) with a scale rate of either 2 or 4. The deconvolutional layer has a kernel size of (3, 3), stride of (2, 2) or (4, 4) depending on the scale rate, and padding of (1, 1) to maintain the spatial dimensions. The convolutional weight has the same shape as the gestures $G \in \mathbb{R}^{12 \times 40 \times 40}$, where the window size is 200ms. $f_{trans,\theta}$ is a 4-layer transformer encoder with input dimension 12, hidden size 96, and output dimension 12. Sinusoidal positional encoding is added to the input of each transformer layer.

### A.11.4 CSA

The flow $f_\theta^G$ is a glow [96] that takes input size 40 and output size 64, which is the phoneme embedding size. For $\mathcal{L}_{phn}$, we predict CMU phoneme targets [102] from MFA or simulation. Note that we use an offline IPA2CMU dictionary to convert IPA to CMU. `https://github.com/margonaut/CMU-to-IPA-Converter`. We use the same text (phoneme encoder) as [119], but with output embeddings size 64. The transition probability $p_\theta(C_i|C_j)$ is simply a (64, 64) linear layer with sigmoid activation.

| Component | Architecture | Details |
|---|---|---|
| GLOW Model ($f_\theta^G$) | Invertible Flow | Input size: 40, Output size: 64, Flow steps: 12 |
| | Actnorm Layer | Scale $s \in \mathbb{R}^{40}$, Bias $b \in \mathbb{R}^{40}$ |
| | Invertible 1x1 Convolution | Weight matrix $W \in \mathbb{R}^{40 \times 40}$ |
| | Affine Coupling Layer | Split input into two parts of size 20 Affine transformation network: 2 FC layers, 64 hidden units, ReLU |

Table 7: Detailed GLOW model architecture

### A.11.5 Language Modeling

In Fig.A.8, we use the same text encoder as[24]. To compute the time information, we use a frame rate of 50Hz and provide it at the word level (for each $C_i$ in the reference text). This time information is then passed to the same text encoder. The embedding sizes are all 4090 [103, 24]. We follow [24] by using a rank of 8 and $\alpha = 16$ in LoRA [105]. All other settings remain the same.

Note that for CSA alignments, we concatenate each $\tau_i$ with its corresponding word $C_i$ alignments, resulting in a 128-dimensional vector. Another encoder, a one-layer MLP (128-4096), maps CSA embeddings into the textual space.

We also have an additional prompt to summarize the actual pronunciation (word, phoneme) and time. The prompt we are using is:

*Given the following text, extract the dysfluent words, the type of dysfluency, and the time of occurrence. Return the result as a list of triples where each triple contains (word, type of dysfluency, time).*

## A.12 Training Configurations

In Eq. 2, $\tau = 2$.

In Eq. 4, $a = b = 1, m_{row} = 3$.

In Eq. 6 and Eq. 7, we simply set $K_1 = K_2 = 1$.

In Eq. 8, $\lambda_1 = \lambda_2 = \lambda_3 = 1$.

In Eq. 12 and Eq. 13, $\delta = 0.9$.

We first separately train the gestural VAE (Eq.8). Subsequently, we train the CSA (Eq.14), followed by training with $\mathcal{L}_{\text{LAN}}$. Finally, we retrain the entire model in an end-to-end manner. For each step, we use the Adam optimizer and decay the learning rate from 0.001 at a rate of 0.9 every 10 steps until convergence. The training is conducted using two A6000 GPUs.

For the VAE and language modeling steps, it takes 40 hours to complete the entire Libri-Dys training. CSA training only takes 5 hours to converge, where only the linear layer transition probability $p_\theta(C_i|C_j)$ is trainable. The same training duration applies to SALMONN [27] and LTU-AS [24] with fine-tuning. Note that we used pretrained models for SALMONN and LTU-AS.

## A.13 Speaker-dependent Behavioral Modeling

Speech dysfluency modeling is fundamentally a clinical problem and, consequently, a speaker-dependent issue. While we have not conducted per-speaker analysis at this juncture, our future research will explore both speaker-dependent and speaker-independent representations [144–148] for clinical analysis. To address potential ethical concerns, we have implemented essential voice anonymization techniques [149] in our processing of disordered speech.

## A.14 Discussion about Concurrent works

As concurrent work, YOLO-Stutter [113] approaches dysfluency modeling as an object detection problem. The authors utilize a simulated corpus and output dysfluency type and timing information. Stutter-Solver [114] further extends this approach, employing a similar pipeline for cross-lingual (English-Chinese) joint simulation and prediction. Notably, Stutter-Solver outperformed H-UDM [2]. Another recent publication, Time-and-Tokens [115], treats the problem as automatic speech recognition, mapping each dysfluency to a token and achieving performance comparable to YOLO-Stutter [113]. Our model primarily emphasizes scalability and user-friendly interface design. Additionally, it establishes a foundation for future researchers to explore in-context learning capabilities. We intend to conduct comparisons with these aforementioned works in future research.

