# OpenReview forum: "SSDM: Scalable Speech Dysfluency Modeling"
_NeurIPS.cc/2024/Conference — NeurIPS 2024 poster_

### Official Review · Reviewer_oU79 · 2024-07-08

**Soundness:** 4
**Presentation:** 4
**Contribution:** 4
**Rating:** 9
**Confidence:** 4

**Summary:**

This is an extraordinarily well-written paper that addresses speech disfluency modeling. Given a recording of a person’s speech and the reference text transcription, the proposed model named SSDM outputs a natural text explanation of the pronunciation problems in specific words and sounds in the recording. This is an extremely challenging task in the speech domain given the sparsity of annotated data and the high cost of producing accurate annotations. The paper as a whole is very novel. I will highlight the main novelties below.

The authors propose a cascade of modeling approaches, drawing upon the learnings from representation learning (VAE, self-distillation), TTS (duration prediction, glow), ASR (CTC, alignment), and self-supervised modeling (acoustic encoders, pre-trained LLMs). It is clear that the authors have put significant effort into developing an interpretable method, which is commendable. They represent the speech signal as a sequence of “gestures” that correspond to articulatory movements, for which pre-trained encoders, VAE, and self-distillation are critical to learn. In order to align the acoustic and text representations well, they find it is necessary to leverage local sequence alignment approaches and they extend the popular CTC objective to perform approximate longest common subsequence alignment in the embedding space. Then, the aligned acoustic-text sequences are provided as the input to LLaMa that is fine-tuned with LoRA for the task of disfluency description. The LLM part of the paper actually acts as the icing on the cake; the main contributions are found in the representation learning across audio, text, and gesture domains, and the alignment method.

In order to train the model, the authors propose a synthetic data creation approach to generate accurately-labeled disfluent speech with TTS. The resulting SSDM model displays state-of-the-art performance in VCTK++ (accented English), Libri-Dys (the synthetic dataset), and nfvPPA dataset which contains real disordered speech, outperforming multiple baselines (LTU-AS, SALMONN, ChatGPT).

**Strengths:**

* Important problem with a big potential for impact that is notoriously difficult to tackle with existing approaches. The authors achieved strong results on the presented benchmarks, which are suitable for evaluating their model.
* Clever application of synthetic data to a task which is known to suffer from a lack of data availability. This choice is very well motivated by the authors who point out that largely dominant alignment-based method suffer from the assumptions about monotonicity and reference text correctness. It is evident the synthetic data creation benefitted heavily from expert knowledge of phonetics and speech production. Bonus points for releasing the dataset as open-source.
* Very well motivated application of monotonic aligners used to identify subsequences instead of individual symbols, and the choice of LCS, further developed by the authors into CCS.
* The appendices contain plentiful information about the details of their approach and experimental setup, but the main body of the paper is written such that it is not necessary to jump into any of the appendices to understand it.

**Weaknesses:**

I couldn’t find any major weakness in the paper, but it could benefit from some revision (see detailed comments).

As for minor weaknesses:
* The model seems to be trained on 100% of synthetic data. It is unclear to what extend it overfits to the specific TTS model instance used to curate Libri-Dys. This question is partially answered by the evaluation on nfvPPA dataset where some degradation of performance is observed compared to Libri-Dys test data.
* As the authors note, the LLM component seems under-utilized. It is unclear whether the LLM overfits to the answer patterns in Libri-Dys dataset. It would be interesting to check to what extent the LLM was able to preserve its original in-context learning and reasoning capabilities to enable the user to have an in-depth discussion about their disfluency patterns.

* p.5, l.163, typo in “variationa”
* p.5, l. 172, was “i” supposed be present in the formula after “we seek an aligner: …”?
* p.5, incosistent naming L_dist vs L_dis
* eq. 10 the expectation over C and tau is missing a comma after the approximation is introduced
* Figure 4 should have at least a rudimentary description of each of the three presented blocks.
* p.7 l.228, should be Eq. 13 not Eq. 2.1
* sec 3.3, I got a bit lost in the explanation of how LCS can be implicitly added to CTC. I think it would be easier to grasp for the reader if the authors explicitly marked which terms in the equations are added/modified vs vanilla CTC objective. I also suggest to explicitly discuss what is the LCS constraint (p.7, l. 217).
* p.9, l.322-323, elaborate what do you mean by “We suspect this is due to the granularity of tokens“
* The references [4-16] have placeholders instead of proper bibliographic information
* The last appendix: “SALMOON” should be “SALMONN”

**Questions:**

* Following up on the issue of using 100% synthetic data, would it be possible to leverage SSDM for weak labeling of data in the wild to develop a non-synthetic dataset? How reliable would SSDM be for this task in author's opinion? I'm asking only for an assessment and not for an experiment, as I realize that's easily a separate follow-up research material.
* A loose thought, but perhaps the system developed by the authors may be used to evaluate TTS outputs, given that some popular classes of TTS models (e.g. decoder-only approaches) suffer from non-monotonic alignment issues. Do the authors anticipate any risk related to overfitting to TTS used for generating Libri-Dys?

**Limitations:**

The authors present a fair assessment of the model's limitations.

---

> ### Author Rebuttal · Authors · 2024-08-05
>
> We really appreciate the time and effort put into giving such in-depth comments.
>
> * Yes. Our model was trained entirely on synthetic data. We chose high-quality open-source multi-speaker TTS models like VITS and StyleTTS2 to mitigate potential overfitting, though we lack concrete evidence of overfitting to specific TTS models. We did tests using VITS on VCTK data and they showed similar results, with slightly lower detection accuracy but still a significant gap between simulated and nfvPPA data. We may have overfitted to fluent TTS models, as no large-scale dysfluent TTS model currently exists.
>
>   * Regarding the questions, we think those are great approaches to try. We all know that current multi-speaker TTS models are trained within a limited set of speakers. We do see efforts in scaling TTS, such as Voicebox, SeedTTS, etc., that were trained on a wider range of speakers. However, the scaling effort is still limited, and those pretrained models are not planned to be open-sourced at this moment. So for a stronger and broader scaling plan, using SSDM on speech in the wild to generate pseudo labels should be a powerful and effective way for further scaling planning. Thank you so much for this insightful suggestion.
>
>   * We strongly agree that SSDM can be used for TTS output evaluation. We can think of two ways to evaluate it: First, for the normal fluent TTS task, we would expect fewer dysfluencies to be detected given ground truth text and speech as input, with our "pronunciation prompt" as additional input. Second, we can also check the CSA alignments output, and we would expect the CSA alignments to be as monotonic as possible.
>
>   * We believe that the evaluation results also depend on the TTS method used for simulation. Popular open-source TTS methods such as StyleTTS2 or VITS are non-AR based methods, which might implicitly inject more monotonicity for speech-text alignments. This might have double-sided effects: (1) The simulated Libri-Dys might not be that natural because of the forced monotonic effect that was naturally encoded. (2) There might also be bias in that our SSDM will be naturally monotonically affected, so some non-monotonic outputs from AR methods might not be affected. These are our assumptions. We should also try AR-based methods to generate simulation data, which should train our SSDM to then evaluate TTS outputs from both AR and non-AR based methods. We think this is a pretty interesting research direction.
>
> * Regarding our CSA alignments: The intuition is that vanilla CTC treats each speech frame - text pair $y^{i,j}$ equally $\forall i,j$, or CTC treats each alignment equally. However, LCS only cares about the "first element" for each sub-sequence and will not take the remaining elements into the loss objective, as shown in Fig. 3. So our purpose is to create some inductive bias to make $y^{i,j}$ not equally considered for different $i,j$.
>
>   * The leftmost sub-figure in Figure 4 gives an algorithm example. Assume we have a true alignment where text $C_2$ is aligned to $y^{2,2}, y^{3,2}, y^{4,2}$. In this example, the emission probability for $y^{3,2}$ and $y^{4,2}$ is actually not important since they are not considered in the loss objective in the off-line LCS algorithm. So there will be two changes in comparison to vanilla CTC: The transition probability should ideally happen between $y^{5,3}$ and $y^{2,2}$ instead of $y^{4,2}$, so $y^{3,2}, y^{4,2}$ are skipped and are called transition skip. However, it is hard for us to really do such a skip operation in the coding. We performed another slack, where we can just make $y^{4,2} = y^{3,2} = y^{2,2}$ so that we still model transition probability between $y^{5,3}$ and $y^{4,2}$. This is what emission copy means.
>
>   * Then let's look at the forward algorithm in the middle part of Figure 4. We have $\alpha^{i,j}$ primarily determined by $\alpha^{i-1,j}$ and $\alpha^{i-1,j-1}$. $\alpha^{i-1,j} -> \alpha^{i,j}$ denotes emission copy. Here we would assume $\alpha^{i-1,j-1}$ has already copied $\alpha^{i-2,j-1}, \alpha^{i-3,j-1},...$, and the transition between $\alpha^{i-1,j-1}$ and $\alpha^{i,j}$ will be able to represent any transition between $\alpha^{i-k,j-1}$ and $\alpha^{i,j}$. So in this method, LCS is recursively encoded. Then we still take into consideration the other text $\alpha^{i-1,j-k}$ with a shrink factor, as this implicitly captures some dysfluencies such as missing. The approximation algorithm is developed primarily based on heuristic analysis, and there is no strict proof of why this approximation is truly LCS-aware. However, our experiments may have already proven this assumption. We are still working on the theoretical aspects to make it more solid. We will make this clearer and add more captions for Figure 4 in the updated work.
>
> * We are still investigating the LLM part. We suspect that LLaMA might be treating this task as a "phoneme transcription" task instead of understanding, since the alignments from CSA already capture highly dysfluency-aware information. LLaMA might just be performing transcription on that. However, we are conducting experiments to do a more in-depth check as you suggested, focusing on in-context learning and reasoning abilities. We also suspect that this may be "due to the granularity of tokens" since LLMs were trained at word or token level (usually corresponding to many frames for speech), while phonemes are at a finer-grained level. So perhaps SSDM would benefit more from a phoneme-level or other hierarchical LLaMA.
>
> * In p.5, l.172, we double-checked and believe there are some symbol issues. We believe there is no "i" since "i" is defined in Eq. 9. The alignment function should be $\gamma: {C_1,C_2,...C_L} \rightarrow \mathcal{P}(\tau_1,...\tau_t')$, which means we map $C_i$ to a subset of the sequence $\tau_1,...\tau_t'$, where $i=1,2..,L$.
>
> * All other typos you proposed are valid, and we will correct them!

---

> > ### Comment · Reviewer_oU79 · 2024-08-12
> >
> > Thank you for your rebuttal. I am satisfied with most of the answers. Regarding the question about the ability to leverage SSDM for weak labeling, I was more asking whether you believe, based on your existing experiments and experience with it, whether SSDM is robust enough to actually succeed in creating a high quality weakly labeled dataset. I realize without actually trying (and possibly iterating many times) it is difficult to make a definitive statement, but any intuitions, expected outcomes, known points of failure, etc. would be interesting to learn. I believe such discussion would make for a good "future work" addition in the paper.

---

> > > ### Author Response · Authors · 2024-08-12
> > >
> > > Yeah, it makes perfect sense to have some practical experimental attempts, which we are trying as ongoing efforts. Really appreciate the valuable suggestions!

---

### Official Review · Reviewer_px45 · 2024-07-10

**Soundness:** 2
**Presentation:** 2
**Contribution:** 2
**Rating:** 3
**Confidence:** 4

**Summary:**

This paper proposes an approach on understanding speech with disfluency.

**Strengths:**

The proposed method is sound.

**Weaknesses:**

This work is an application work focused on a very narrow domain in the speech area. It doesn't seem to have sufficient interest for the audience of NeurIPS. It probably fits better for another speech-focused venue such as Interspeech or ICASSP.

Updated: There is also concerns on the correctness of the evaluation during the author-reviewer discussion.

**Questions:**

No.

**Limitations:**

Yes.

---

> ### Comment · Reviewer_px45 · 2024-08-01
>
> Sorry for the brevity in the previous review.
>
> My main concern on this submission is: it is an application work focused on a very specific task on a narrow domain, which doesn't seem to have sufficient interest for the general audience of NeurIPS.
>
> According to this paper (Sec. 1 and Sec. 2.1), speech dysfluency modeling is a task that detects both the types and the timing of disfluency in speech recordings given the reference text. By this definition, it is a very specific and narrow domain task, with forced alignment as the core technique.
>
> All the key related works on this task as referenced by this paper in Sec. 1 [1, 2, 32, 33, 34, 35, 36, 27, 38, 39, 40, 41, 42, 43, 44, 45, 46, 47, 48, 49, 50] and [82] in Sec 2.1, including the "current state-of-the-art solutions [1, 2]", were published in various domain specific venues, notably in ICASSP and Interspeech.
>
> Because this work is quite task-specific, I personally don't have an expertise for a thorough assessment for its technical contribution on this task.
>
> That said, in addition to the main concern, I do have a few technical questions:
>
> 1) The proposed method uses an LLM (Llama) as the last components, which outputs in free form text. How were the numeric-based evaluation (F1, MS, etc.) done with the free-form text response?
>
> 2) According to Figure 2, the proposed method predicts only a single timestamp as the timing for each disfluency instance. What does this timestamp represent, starting, ending, or something else? Either way, without being a time range, how does IoU (Intersection over Union)-based Matching Score calculated in the evaluations?
>
> 3) In Sec 6.5 (and Table 3), the paper claims the proposed method as "State-of-the-art Dysfluency Detection". However, it's not compared to either of the "current state-of-the-art solutions [1, 2]". Although there is a comparison to [2] in Table 2, but Table 2 only include results on simulated datasets, without using the real disfluency dataset (nfvPPA). Does such evaluation support the claim?
>
> 4) In Table 3, ChatGPT is used as a baseline. Isn't ChatGPT a text-based model? How does it work for detecting disfluency in speech?

---

> > ### Author Response · Authors · 2024-08-07
> > **Response to follow-up- questions**
> >
> > We have attempted to address the main concerns in rebuttal part. Thanks so much for your follow-up questions. Here are our explanations for these questions. (We attempted to submit it during the rebuttal period, but due to technical issues, we were unable to do so until now. )
> >
> > 1. LLMs truly give a general diagnosis or feedback. We applied pre-designed prompt engineering to obtain time and type information. This was mentioned in Appendix 11.5. However, during inference, we also perform human intervention to check the actual outputs after prompt engineering for 100% controllability.
> >
> > 2. This is a good question. I think the previous paper [1] provides detailed explanations for those metrics. We did not detail them again here. However, we do have a visualization at the end of our demo page (link is in the abstract). We will make this more clearly visible in either the main paper or appendix in the future. Thank you!
> >
> > 3. We would like to stress this question more. It is true that we have been working on inference for disordered speech with baseline work [2], which is presumably better than other SLMs such as SALMONN, etc. However, the data is from the clinical domain and is extremely sensitive. We temporarily lost access to some of that data for some reasons. Thus, we only evaluated about 2 speakers, not the full set by the time of submission. The accuracy (F1, MS) is much lower than ours. Unfortunately, we did not report the full results from [2]. After paper submission, we regained access and now have the exact numbers. For H-UDM in [2], it is 55.6% F1 score and 46.3 for MS. We will update these numbers in the revised work. Thank you so much for pointing out this question!
> >
> > 4. This is also a question from another reviewer, so we can use the same clarification here: ChatGPT was initially proposed as a language model only. However, in 2023, it also included speech I/O and became available (only) in mobile apps. So we now refer to ChatGPT as a general assistant that can process multiple modalities. We used the iPhone ChatGPT app, the 4-o version, and conducted the inference experiments in May this year. However, we also note that the 4-o version with speech I/O currently available in the app is not the true 4-o speech demo they presented recently.

---

> > > ### Comment · Reviewer_px45 · 2024-08-12
> > >
> > > Thank authors for the response.
> > >
> > > Re: Relevance
> > >
> > > The authors agreed that this paper was on a narrow research domain, while it did not mean that this domain was not important. My concern here is on the relevance of this paper to the general audience of NeurIPS. The authors' response did not address this concern.
> > >
> > > The author claimed this paper to be a "LeNet moment". However, I honestly do not see a connection. If the authors do believe so, please add a section in the paper to explain the potential impact of this work beyond the domain of dysfluent speech modeling and beyond the domain of speech.
> > >
> > > Re: Techincal questions
> > >
> > > Q1: 1) This is an important part of the evaluation protocol so that it should be explained in the main context, instead of in appendix. 2) How much is the impact of the human correction? Is it even a fair evaluation, while the reported results include human correction?
> > >
> > > Q2: This question is not address. [1] predicts time boundaries for each disfluency instance, so that they could calculate an intersection-over-union ratio against the reference time ranges. This paper predicts a single timestamp rather than a time range for each instance, then how was intersection-over-union calculated?

---

> > > > ### Author Response · Authors · 2024-08-12
> > > >
> > > > Thank you for your comments. We feel there are still some misunderstandings that we'd like to clarify and hopefully we can work together to make it clear.
> > > >
> > > > * Regarding your primary concern
> > > >
> > > > While we indeed mentioned that not many people are working in this direction, it doesn't mean it's a narrow research domain. We used "narrow" in quotes. Since you also admitted "it did not mean that this domain was not important," an important domain should expect to receive enough attention at NeurIPS. We think it's normal that for some domains in NeurIPS papers, many people are working on them, while for others, fewer people are involved. So we might not evaluate it based on the number of people working in that area, but rather on how important it is, which you somewhat admitted. Our point is that as long as it's an important area, it should be eligible for NeurIPS submission. We would greatly appreciate feedback on our actual technical contributions (gesture, VAE, CSA, etc). Additionally, whether a paper fits the NeurIPS venue or not should have been reflected to area chairs before the review starts, not at this moment. So we believe you've already realized the importance of this work to this extent and you are just concerning the number of people working on this area. If you agree with this, we believe your primary concern might be addressed. Feel free to discuss this further. We suggest looking at the potential impact of this work rather than the number of people working on it.
> > > >
> > > > * Regarding your secondary concerns:
> > > >
> > > > We call it a "LeNet" moment because this is an important area where there's no large-scale, high-quality data and no effective learning framework. We're happy to revise this. Also, in the first paragraph of the introduction, we indeed mentioned the importance of this paper beyond the domain of speech (l.15 - l.23 for its huge clinical and business impact). Let us know your suggestions, and we're happy to revise, remove, or add more.
> > > >
> > > > * Regarding minor questions:
> > > >
> > > > Q1. (1) We admit that the evaluation metric isn't mentioned much in the main context, primarily due to space constraints. This is a weakness we acknowledge. We'll add it to the main context in the updated work.
> > > >
> > > > Q1. (2) By human correction, we only fixed some grammatical presentation issues. For example, if the output says "3.40s," we would only keep its numerical part "3.40". So the "results" are not corrected by humans.
> > > >
> > > > Q2. This is a good question. We definitely predict a boundary. However, when start and end times almost overlap, we combine them for visualization. This works especially for missing, deletion, or insertion cases. We output a region like "2.20s-2.21s". We also have region-wise targets. So we still use the same IoU metric. Sorry for the confusion; these are details that weren't included. We'll include them in the main context and make it clearer.
> > > >
> > > > Please let us know whether your primary, secondary, or all concerns have been addressed. We're happy to discuss further if you have more questions.

---

> ### Author Rebuttal · Authors · 2024-08-05
>
> Dear Reviewer,
>
> We understand that confusion or misunderstanding may arise from the definition of this research area. We believe we can agree that in the speech research community, the majority of people are working on TTS, ASR, or spoken language models (SLMs) due to the breakthrough of generative AI and LLM techniques. Only a few people are working on some rare research areas like our domain: dysfluent speech modeling. Because of this, most of the works we have heard about recently are related to TTS, ASR, LLMs, SLMs, etc., especially for those who have just entered the speech area for research and are excited about it. We believe this is the fundamental reason you might think this is a narrow domain rather than a popular or general one. This is totally understandable and makes sense.
>
>
> However, we would like to explain our motivation further. The fact that not many people are working on this topic certainly indicates it is a "narrow domain," at least at this moment. However, it does not mean it is not important. We also indicate in the abstract that the research of dysfluency modeling is still at its "LeNet moment." So at the time when LeNet was invented, maybe other people also thought it was a narrow domain and not important. That's why Hinton's neural network was not taken seriously until many years later.
>
> Before elaborating on our motivation for this work, we might pose a general question: Why are so many people working on popular areas like TTS or SLMs in the speech domain? We think it's fundamentally driven by the needs of the world, or more specifically, the market size. Because there is a huge need for some techniques, money will follow, and many people are needed in this area. Usually, people working together will create much more impactful work, such as those highly influential speech works we cited in the related work section.
>
> If this is true, we have also done early market investigation showing that the market size for dysfluent speech (speech therapy and spoken language learning) is also substantial. We already reported the numbers in the introduction. The exact figure for the speech therapy market value (6.93B) is even larger than the TTS market size (3.45B) and is comparable to ASR. This means that, in terms of needs and market size, dysfluent speech modeling is as important as other areas such as LLMs, TTS, SLMs, ASR, etc. Also, we reported at the beginning of the introduction that 1 in 10 kids globally suffer from dyslexia. What does this mean? It means if I have a child in the future, there's a 1/10 probability that my child will also suffer from dyslexia. Since there is not an effective AI framework, I would absolutely encourage my child to try this SSDM to get a diagnosis. This 1/10 probability applies to all people globally. In this case, maybe this is even a more important area than popular areas such as TTS or ASR. We're not trying to compare which area is more important, which would be meaningless. We just mention that this is not a narrow domain and instead will become an increasingly important one.
>
> But you might further ask, since it's such an important area, why aren't more people working on it? We mentioned this in the introduction and related work sections, as well as in the demo on the first page. The current large-scale SLMs such as LTU-AS, SALMONN, ChatGPT have limited ability to tackle this problem. Although we mentioned that, technically, dysfluent modeling is a spoken language understanding problem that could be classified under either ASR or SLMs, the work including [1,2] and our method indicate that the efforts needed to bridge the gap between ASR/SLMs and dysfluent modeling are much greater. It is a super difficult problem. Because of this difficulty, we personally think dysfluent speech modeling should be separated as an independent research area. Even in some speech conferences such as Interspeech/ICASSP/SLT/ASRU, this area has not been defined. So when we were suggested to submit this work to Interspeech/ICASSP, there might still not be a suitable venue for us.
> Regarding the specific point that the reviewer has raised, we checked the official NeurIPS requirements again and found:
>
> "The Thirty-Eighth Annual Conference on Neural Information Processing Systems (NeurIPS 2024) is an interdisciplinary conference that brings together researchers in machine learning, neuroscience, statistics, optimization, computer vision, natural language processing, life sciences, natural sciences, social sciences, and other adjacent fields. We invite submissions presenting new and original research on topics including but not limited to the following: Applications (e.g., vision, language, speech and audio, Creative AI)."
>
> So we would think the NeurIPs still encourages the speech application work submission. But if this is the core concern and still not clear, we would also love to invite the program chairs or area chairs to look through our work again and to analyze our qualifications.
>
> We have clarified our points regarding the reviewer's primary concerns. We understand that the confusion happens because there are indeed not too many people working in this area. However, we are making efforts to make this area more well-known in the future and to attract a wider audience. So feel free to discuss this further. If possible, we would also love to invite you to help announce this paper to a broader audience in the future to benefit more patients with speech disorders.

---

> ### Author Response · Authors · 2024-08-13
>
> Dear reviewer:
>
> Thank you for your attention to our research. We greatly respect your perspective on our research domain, which we acknowledge is somewhat narrow. However, we cannot construct a meaningful rebuttal without understanding your specific concerns. We would appreciate clarification on any technical flaws you've identified, such as issues with our experimental design or methodology. We are eager to address these concerns or any confusion you might have. We also welcome any suggestions you might offer.
> We are happy to provide a more detailed explanation of our methods and to re-explain our work and its importance using clear, accessible language. We will explain our research step-by-step, from background to technical details, to enhance your understanding.
>
> * Approximately 1 in 10 people suffer from speech disorders such as aphasia or dyslexia.
>
> * When people suffer from speech disorders, they typically seek treatment at hospitals. However, the cost is often prohibitive, usually exceeding $1,000 for a single diagnosis, which can take over 5 hours. Additionally, there is a shortage of doctors and speech-language pathologists (SLPs). Consequently, such costs are often unaffordable for low-income families. Even in countries where the government covers these fees, treatment is time-consuming, and many people lack access due to the scarcity of specialists.
>
> * This raises the question: Can AI automate this process? AI has the potential to provide an efficient and more affordable solution.
> For AI to automate speech disorder diagnosis, it needs to perform deep transcription of speech, including dysfluencies such as insertions, omissions, replacements, stutters, prolongations, and phonetic errors. Accurate timestamping is also crucial. SLPs have informed us that these are the official diagnostic procedures.
>
> * Here's an example of what an AI model should output: Given an impaired speech sample, we expect our system to produce: "Y-You [block] wi-[stutter]-sh [missing] know all [prolongation] about my grandfather". In contrast, state-of-the-art models like Whisper, ChatGPT, or Salmonn typically produce a perfect transcription: "You wish to know all about my grandfather", which is not useful for our purposes. This demonstrates that current large-scale Automatic Speech Recognition (ASR) models show limited performance on these tasks.
>
> * To underscore the importance of this research: Recall that a single treatment session costs about \\$1,000 and takes approximately 5 hours. Our AI solution could potentially save both patients and healthcare providers (or governments) \\$1,000 and 5 hours per session. Given that 1 in 10 people have disordered speech, the market size is substantial. To our knowledge, AI exploration in this area is limited. Therefore, our work has broad impact, including both clinical and business implications.
>
> Now, let's discuss the technical aspects of our work. We will introduce these step-by-step, and please don't hesitate to ask for clarification.
>
> Our pipeline has two inputs: dysfluent speech and either golden truth text or a prompt.
> * We obtain speech representations from dysfluent speech.
>     * 2.1. While we could use traditional speech representations such as HuBERT, we found that our gestural scores are more scalable for this task, as demonstrated in Table 1. Therefore, we employ gestural scores instead of traditional SSL units.
>     * 2.2. To obtain gestural scores, we first need articulatory kinematics data. We use Acoustic to Articulatory Inversion (AAI) to derive articulatory data X from speech. We then developed a VAE framework to generate gestural scores H from X. Gestural scores are sparse and duration-aware, so we developed a duration predictor, intensity predictor, and sparse sampler. We're happy to discuss any of these steps in more detail if you have questions.
>
> * Connectionist Subsequence Alignment (CSA): We discovered that the Longest Common Subsequence (LCS) algorithm naturally captures speech dysfluencies, as shown in Figure 3. For the ground truth phoneme "AH", it aligns to the speech "AH AH ER AH", indicating a repetition of "AH" and an insertion of "ER". Based on this discovery, we developed a stochastic and differentiable LCS, named CSA, based on the vanilla CTC algorithm to provide dysfluency-aware alignments.
>
> * CSA takes gestural scores H and text as input and outputs an alignment. This alignment is then input into LLaMA with pre-designed instructions to achieve end-to-end learning.
>
>  * Due to the lack of large-scale data, we developed a large-scale, TTS-based simulation dataset called Libridys, comprising about 4,000 hours of speech. We trained our entire model on Libridys. We then performed evaluations on Libridys, VCTK++, and real disordered speech data (nfvPPA).
>
> We look forward to your feedback and are happy to provide further explanations or clarifications as needed. Please let us what step might confuse you.

---

> > ### Comment · Reviewer_px45 · 2024-08-13
> >
> > Thank authors for the additional responses. Based on these responses, I have a concern on the technical correctness of the evaluation and the clarity of the writing on them. Both Fig 1 and Sec A.11.5 show that the proposed method predicts a single timestamp for each disfluency instance, which would be insufficient for calculating an intersection-over-union ratio. The authors' first response didn't address the question; the second response stated that they outputted a time region instead of a single timestamp, which is different from the paper.
> >
> > I updated my ratings based on the technical concerns.

---

> > > ### Author Response · Authors · 2024-08-13
> > >
> > > Dear Reviewer,
> > >
> > > Thank you for the questions you proposed. We acknowledge that there are some aspects we did not clearly explain in the paper. However, there are no technical errors, and we will do our best to clarify these points for you.
> > >
> > > * In our original annotations, the time annotation for a specific type of dysfluency is a region, not a timestamp. We have open-sourced our data in our manuscript, and you can verify this manually. Note that for some dysfluencies, such as missing words, the start and end times might be the same or very close. For example, it might be 2:00s-2:00s or 2:00s-2:06s. To reduce confusion, here is an exact sample of our JSON format labeling: {"word": "stella", "start": 0.005, "end": 2.005, "phoneme": "t", "start": 1.125, "end": 1.175, "type": "replace", ...}
> > >
> > > * Figure 12's "annotation part" might be confusing as we only intended to show readers where the dysfluency starts. Introducing additional information like duration (e.g., "2:00s-2:06s") might make it more confusing since the current system is already quite complex. You may have noticed that we have time-embedding per frame, so it is definitely region-wise modeling. The figure is an illustration of annotation, not the actual annotation. For real annotations, please refer to our open-sourced data. However, if this causes further confusion, we will update Figure 12 in our future work. We welcome any advice on improving it.
> > >
> > > * The actual output from the model is a region, not a timestamp. We use this region to compute the MS score, which is consistent with [1,2]. Note that for some specific types of dysfluencies, such as missing words, the output is exactly a timestamp or pretty short region. In these cases, we still report exact matching during MS computation, replacing IoU. However, it is essentially still IoU. (duration-free F1)
> > >
> > > * We have received suggestions from doctors that the starting time or end time is usually more important. Therefore, we filter out those regions and only keep the starting timestamp as the final visualization for friendly user interface. However, region-wise prediction and evaluation are essential. And if you think keeping region information during visualization stage is critical, we will definitely do that. There is no conflict between these approaches.
> > >
> > > In conclusion, we do not have the technical errors you mentioned. The misunderstanding might stem from our lack of detail or clarity in the paper. We will definitely add these experimental details in the updated work.
> > >
> > > We sincerely hope this clarifies your technical concerns. Feel free to discuss!

---

### Official Review · Reviewer_nbR5 · 2024-07-12

**Soundness:** 3
**Presentation:** 3
**Contribution:** 3
**Rating:** 7
**Confidence:** 4

**Summary:**

This paper looks at the problem of disfluency events in speech. Their core contributions are:
* Instead of learning dysfluencies that rely on high-compute SSL models, they characterize speech representations using a type of gestural modeling baed on kinematic/articulatory movements. They rely on a pre-trained acoustic-toarticulatory inversion (AAI) model that regresses from speech to EMA features. They align information from the gestures (using the AAI model) with this and self-distillation.
* They describe a forced aligner (and connectionist subsequence aligner) to align dysfluencies within their sequences.
* They introduce a synthetic dataset built on LibriTTS with almost 4k hours worth of data.

The results across tasks including disfluency detection and (dysfluent) text to speech are compelling.

**Strengths:**

Overall this is a really interesting and deep paper on a topic that doesn't get that much attention.
* The use of acoustic-to-articulatory inversion features is interesting given how they're used in a scalable/universal manner.
* The various connection to LLMs is relevant and interesting, even if they aren't strictly necessary
* The dataset looks like it could be really valuable for the community
* The diagrams do a nice job of outlining what is an otherwise complicated system
* I thought the additional TTS experiments in the appendix were really compelling. I wonder why they weren't included in the main part of the paper.

**Weaknesses:**

My biggest gripe is that the overall system is very complicated, which is not a problem in itself, but it's challenging to understand how everything fits together and how why certain design decisions were made. There is SO much work in this paper, but it can be very hard to follow especially if you include the appendices. Many of the technical pieces are described in detail, but don't have proper motivation. For example, why do you need self-distillation? I have a sense of how it works but not why it's necessary. I found that there are many concepts like this.

Similarly, some terms are not explained. For example, the metric dPER should have a description -- I see the reference but you need at least a brief explanation in this paper.

**Questions:**

There are results in Table 3 corresponding to "ChatGPT." This isn't explained anywhere in the paper. It's unclear how a text-only model like ChatGPT could be used in this setting.

**Limitations:**

Yes they do a reasonable job.

---

> ### Author Rebuttal · Authors · 2024-08-05
>
> Thanks reviewer for the thoughtful summarization and comments.
>
> * Regarding the TTS question proposed in Strengths: We summarize that we performed textual dysfluency editing and simply input that text into a pretrained StyleTTS2 model, which is one of the open-sourced state-of-the-art TTS works. Our contribution is primarily in textual engineering, and we did not tune the model itself. Therefore, we believe this part of the contribution, at least technically, might not be novel enough to be considered one of our main contributions. We did observe some unnatural simulations, such as robot-like noise in stutter simulation, which we believe requires a learning method to overcome. For example, we were considering developing a GAN objective and fine-tuning the StyleTTS2 framework to address these issues. However, we have not yet accomplished this. Once we achieve a new technical-level breakthrough in TTS, we would be happy to list that as a core contribution. Due to page limits and the existence of other technical contributions (VAE, CSA, etc.), we do not have much space to elaborate further here. At this moment, we believe the core contribution might still be the open-sourced data. We hope to develop an updated TTS framework in our future work, and any suggestions are welcome!
>
> * We acknowledge that our entire pipeline is complex. We attempted to simplify the system design initially but did not achieve the desired performance boost that we now have. As of now, all modules we proposed or utilized play an essential role and cannot be replaced. However, we are continuing our efforts to make SSDM simpler, more transferable, and more deployable. The reason we include so many details in the appendix is that we feel some essential modules, like gestures, are not well-familiarized in the speech domain. We hope that in the future, as people begin to realize the existence and importance of articulatory speech modeling, this article will be naturally simplified.
>
> * Regarding specific questions, we believe we have introduced both motivation and methods in detail. However, some of this information might be included in the appendix due to page limits. For example, regarding self-distillation, there are two motivations we wish to explore:
>
>   * Articulatory kinematics data are sparsely sampled from human articulators. This definitely results in some information loss, including intelligibility loss, pitch, or speaker identity information loss. The advantage is that it is scalable, as proven in Table 1, and interpretable for pronunciation training, as shown in section 6.6. So, how can we take advantage of these benefits while also guaranteeing intelligibility? We have "borrowed" more intelligibility from WavLM via self-distillation. This motivation was briefly discussed in lines 79-84 and detailed in Appendix A.3.
>
>   * It has been evidenced in [1] that gestural score H implicitly captures linguistic boundaries such as phoneme segmentation. We also drew intuition from DINO [2], which showed that self-distillation in visual representations can implicitly enable image object segmentation, a so-called "Emerging property". Following work in speech [3] shows that self-distillation can also implicitly enforce word segmentation and better intelligibility [4]. Based on this intuition, we use self-distillation to enforce linguistic boundaries for gestural scores and to improve the upper bound of intelligibility that WavLM already achieves. These points are discussed in detail in Appendix A.3, where we also list other papers in this area. We mentioned in the main paper on line 159 that you can refer to our Appendix. However, we still feel sorry that due to page limits, we can only list this information in the appendix.
>
> * We admit that dPER was indeed not well explained. Although we cited its reference, we should have explained it in our paper. Thank you for the suggestion!
>
> * Regarding the question, ChatGPT was initially proposed as a language model only. However, in 2023, it also included speech I/O and became available (only) in mobile apps. So we now refer to ChatGPT as a general assistant that can process multiple modalities. We used the iPhone ChatGPT app, the 4-o, and conducted the inference experiments in May this year. However, we also note that the 4-o version with speech I/O that is currently available in the app is not the true 4-o speech demo they presented recently.
>
> [1] "Deep Neural Convolutive Matrix Factorization for Articulatory Representation Decomposition"
>
> [2] "Emerging Properties in Self-Supervised Vision Transformers"
>
> [3] "SD-HuBERT: Sentence-Level Self-Distillation Induces Syllabic Organization in HuBERT"
>
> [4] "DinoSR: Self-Distillation and Online Clustering for Self-supervised Speech Representation Learning"

---

> > ### Comment · Reviewer_nbR5 · 2024-08-10
> >
> > Thanks for the detailed response. For the final version, please include text for the last two bullets describing dPER and the ChatGPT approach as these are both missing from the text.

---

> > > ### Author Response · Authors · 2024-08-12
> > >
> > > Yes sure. Thanks for all the suggestions!

---

### Official Review · Reviewer_YZds · 2024-07-14

**Soundness:** 4
**Presentation:** 3
**Contribution:** 4
**Rating:** 7
**Confidence:** 4

**Summary:**

This paper presents SSDM (Scalable Speech Dysfluency Modeling), a novel approach to modeling speech dysfluency, which is essential for applications in spoken language learning and speech therapy. The authors identify three main challenges in the current state-of-the-art: poor scalability of existing solutions, the absence of a large-scale dysfluency corpus, and the lack of an effective learning framework. To address these issues, SSDM incorporates several key innovations:

1. Articulatory Gestures as Scalable Forced Aligner: The paper proposes using neural articulating gestural scores as scalable representations for dysfluency modeling, grounded in physical laws of speech production.

2. Connectionist Subsequence Aligner (CSA): Introduced as a differentiable and stochastic forced aligner to achieve dysfluency alignment by linking acoustic representations and text with dysfluency-aware alignment.

3. Libri-Dys Dataset: The authors have developed a large-scale simulated dysfluency corpus called Libri-Dys to facilitate further research in the field.

4. End-to-End System with Large Language Models (LLMs): The system leverages the power of large language models for end-to-end learning, aiming to enhance the understanding of human speech nuances.

**Strengths:**

1. SSDM provides a holistic solution by combining articulatory gestures, a new aligner (CSA), a large-scale dataset (Libri-Dys), and leveraging large language models, which could significantly advance the field.
2. The work has direct implications for speech therapy and language learning, which are significant areas with real-world impact, especially for individuals with speech disorders.
3. By open-sourcing the Libri-Dys dataset, the authors contribute to the community by providing a valuable resource for further research and development.

**Weaknesses:**

1. The paper writing is unclear, which makes it quite hard to understand the whole works. For the Figure 2, the author should make it clear which part is used in training and which part is used in inference. Also there should be more explanations on difference of the line and dashed line. Moreover, a detailed overview of the whole system about the training and inference, respectively, is necessary for the complex system like this.

2. From the results in Table 2, it seems that LLAMA is not a necessary part for the system to be effective. I wonder if there is any traditional method on this task, i.e., the output of traditional system is the category label of error type and its duration. Comparison with such systems is also needed.

3. Since the baselines in Section 6.5 uses only speech input, it seems to be an unfair comparison, which explains the large gap between the proposed method and baselines.

**Questions:**

1. Is there any ablation study on the number of gesture number?
2. Is there are any sample or metric showing how well can we recover speech from the articulatory representations.
3. Is mask in Figure 9 necessary?

---

> ### Author Rebuttal · Authors · 2024-08-05
>
> Dear Reviewer:
>
> We are pleased to address your proposed weaknesses and questions.
>
> **For weaknesses**:
>
> 1. We acknowledge that the numerous modules and Figure 2 may be challenging to grasp. The training process involves all modules in Figure 2. For inference, we follow this path: speech -> gestural scores -> CSA alignments + instruction -> LLaMA -> response, removing modules like self-distillation and gestural decoders. Solid lines represent formal I/O processes in neural networks, while dashed lines denote other cases, such as loss functions. In the *general* rebuttal, we also provide additional explanations about our entire pipeline for further review. We will clarify it further in the revised version.
>
> 2. Our results show a small improvement with LLaMA, which we are still investigating. One hypothesis is that LLaMA was trained at the token or word level, whereas SSDM operates at the phoneme level. A phoneme-level LLaMA might yield more significant improvements. We do have a traditional method (as reviewer mentioned, the category label output), which is exactly the H-UDM [2]. H-UDM outputs the type and time of dysfluencies without integrating a "smoothed" language model. Tables 1 and 2 compare H-UDM in terms of scalability and detection effectiveness.
>
> 3. We compared three baselines: LTU-AS, SALMONN, and ChatGPT, which represent state-of-the-art speech understanding models. LTU-AS and SALMONN were trained via instruction tuning with both speech and **text** inputs, like our approach. So we believe it is fair comparison. For ChatGPT, we used the GPT-4o version from the iPhone app with a speech I/O interface, though it's unclear if it still uses Whisper. This comparison is fair given the extensive training data for these models. For a better fair comparison, we also fine-tuned LTU-AS-13B-FT and SALMONN-13B-FT with both speech and text inputs, but the performance boost was limited.
>
> **For questions**:
>
> 1. Previous research [1] explored gesture numbers of 20, 40, 60, and 80, finding that 40 gestures yielded the lowest PER on MNGU0 dataset. While this may not apply to our larger Libri-Dys dataset, we adopted 40 gestures as default, which also aligns with the number of CMU phonemes. Using [1]'s code, we tested higher gesture numbers (100, 200, 1000) on MNGU0. Although intelligibility didn't degrade, interpretability decreased, making it difficult to associate gestures with specific articulatory movements. This interpretability is crucial for providing articulatory-based pronunciation feedback (Sec. 6.6). Thus, we used 40 gestures in our formal experiments. We will explore numbers below 100 in a revision.
>
> 2. There are two types of articulatory representations: a. Articulatory kinematics data X from UAAI; b.Gestural scores H derived from X. Both serve as "representations" of speech and are considered "articulatory representations."
>
>   * Articulatory data X, sparsely sampled from six human articulators, primarily conveys speech content but may miss acoustic details like speaker pitch. Research [4] shows that articulatory kinematics can serve as a universal speech codec. When combined with speaker embedding and signals like pitch, it can fully recover intelligible speech, achieving decent WER, CER, and MOS scores for resynthesized speech (LibriSpeech, VCTK, and AISHELL) as shown in Tables 2 and 3 of [4].  Note that [4] also primarily considers articulatory data as speech content, playing the same role that HuBERT plays in [5] or wav2vec2 plays in [6].
>
>   * Gestural score H from articulatory data X:
> Due to slight information loss in articulatory data, there is also inevitable loss in the gestural score H, as we focus on reconstructing articulatory data X. Our goal is to leverage its scalability and interpretability. To address this, we use self-distillation, allowing H to distill more acoustic information from pretrained WavLM representations. We believe this makes our articulatory representations H nearly fully intelligible. Although we did not directly evaluate H's information retention, we used two implicit metrics:
>
>     * In [3], the original reconstruction loss when recovering X from H is about 20% with 40 gestures, corresponding to a phoneme error rate of 11.02 (7.52 on melspec). This indicates a loss of intelligibility in the gestural score. With our self-distillation method, the reconstruction loss drops to 2.08%, showing nearly full intelligibility. We plan to include more analysis in the updated paper.
>     * Table 1 reports phoneme classification accuracy (F1 score), showing that our gestural score H outperforms HuBERT representations and achieves full intelligibility from speech.
>
> 3. Yes. Gestural scores H differ from traditional speech representations due to physical constraints. [1] defined speech as temporal activations of sparse gestures (articulators), making gestural scores naturally sparse. We applied a sparse constraint for physical consistency. Without the sparse constraint (mask), we still achieved good detection results. However, two points are noteworthy:
>
>   * Scalability is affected: Without the mask, we tested that scalability becomes SF1=0.14 and SF2=0.17, matching HuBERT (Table 1).
>
>   * Interpretability is lost: Without the sparse constraint, gestural scores lose interpretability, making articulatory pronunciation feedback (Sec 6.6) impossible.
>
>     In essence, while removing the mask doesn't significantly impact detection, it affects scalability and eliminates interpretability benefits.
>
> [1] "Articulatory phonology: an overview"
>
> [2] "Towards Hierarchical Spoken Language Dysfluency Modeling"
>
> [3] "Deep Neural Convolutive Matrix Factorization for Articulatory Representation Decomposition"
>
> [4] "Articulatory Encodec: Vocal Tract Kinematics as a Codec for Speech"
>
> [5] "Speech Resynthesis from Discrete Disentangled Self-Supervised Representations"
>
> [6] "Neural Analysis and Synthesis: Reconstructing Speech from Self-Supervised Representations"

---

> ### Author Response · Authors · 2024-08-12
>
> Dear Reviewer:
>
> We again acknowledge that the writing is not as straightforward as it could be due to the complexity of the subject matter, and we are making efforts to improve its visualization (global rebuttal and future work). Apart from that, please let us know if your primary concerns/questions have been addressed. Feel free to discuss or ask any other questions you may have so that we can provide prompt feedback!

---

### Author Rebuttal · Authors · 2024-08-05

Dear Reviewers:

Given the complexity of our system design, we will elaborate more on our pipeline in this general rebuttal.

While we believe the paper's writing is generally clear, we recognize there is room for improvement. Our core method involves generating gestural scores from speech, aligning these scores with text supervision, and then feeding this alignment into LLaMA for training and inference.

We use gestural scores rather than traditional speech features like SSL because they are more scalable for this task. Alignment is necessary because dysfluencies are naturally encoded in speech-text alignments. We incorporate LLaMA to make the entire system end-to-end and to explore whether external LLMs can enhance dysfluency detection. To develop gestural scores, we employed a VAE for expressive posterior modeling and a duration model. We use sparse sampling (mask strategy) to make gestural scores consistent with theoretical support [1]. The reasons for using self-distillation are twofold: (1) articulatory information from AAI suffers from information loss; and (2) self-distillation implicitly derives emergent abilities such as image segmentation and word segmentation, which can improve the semantics of the gestural scores. We believe the overall logic is well-explained in the paper.

However, we admit there are aspects we must improve to enhance clarity, as you suggested. Firstly, Figure 2 contains many modules, and we should clearly differentiate between training and inference steps. Due to page limitations, we did not initially separate these, which unfortunately caused confusion. We would like to clarify further: The training process involves all modules in Figure 2. We have two training paradigms, as mentioned in Sec. 6.2. The first method trains VAE, CSA, and LLaMA separately. VAE training includes gestural posterior modeling, self-distillation, and dysfluency alignment classification, up to dysfluency alignment. The dysfluency alignment uses pre-simulated forced alignment as a target (dashed line). The loss objective is defined in equation 8. We then fix the VAE and use dysfluency alignment (green blocks) and reference text embedding as input to train CSA with the objective defined in equation 14. Next, we fix VAE + CSA and use the output alignment embedding from CSA, along with language instructions and targets (shown in response), to fine-tune LLaMA with LoRA. The second training method involves training the entire system (VAE+CSA+LLaMA) end-to-end, using the separately trained models as initialization. For inference, we follow the path: speech -> gestural scores -> CSA alignments + instruction -> LLaMA -> response. Modules like self-distillation (acoustic adaptor parts), and gestural decoders are no longer needed during inference.

We apologize for any confusion this may have caused. Although these steps are detailed in the paper, we should provide better visualizations for training and inference separately. In our updated paper, we will either simplify Figure 2 and clearly label training and inference steps, or provide more detailed instructions in the appendix. We welcome any advice on this matter. We will provide a detailed caption to explain this or simplify Figure 2 in the updated work.

---

### Decision · Program_Chairs · 2024-09-25

**Decision:**

Accept (poster)

**Comment:**

1 strong accept, 2 acccepts and 1 reject
the reject reviewer wrote very little validation about why rejecting, other than it is better suited for ICASSP or Interspeech. This is not a reason to reject a paper.
The accepts all point to interesting idea and results, as well as a paper being well written.